# CAPO: Conflict-Aware Policy Optimization for Large Language Models

## Abstract

Recent advancements in policy optimization techniques have profoundly improved the reasoning abilities of large language models (LLMs). A pivotal breakthrough lies in sampling a group of responses for each query and adjusting their likelihoods based on the relative advantages of their scores over the group mean. However, substantial conflicts may arise between the aggregated gradient and the individual gradients of the responses, thus diminishing the effectiveness of gradient signals and ultimately hindering the training performance. To address this challenge, we propose **C**onflict-**A**ware **P**olicy **O**ptimization (**CAPO**), a novel and scalable training method that mitigates conflicts through dynamic gradient aggregation. Specifically, CAPO formulates the gradient aggregation step as a *second-order cone program (SOCP)*, which seeks a gradient direction maximizing the alignment with positive-advantage responses, while enforcing constraints to suppress negative-advantage responses. To equip the SOCP with scalability and tractability for LLMs, we significantly reduce the number of variables via the Lagrangian duality and compress the gradient dimension using the Johnson-Lindenstrauss transform. We further show that the dynamic gradient aggregation effectively reduces conflicts without sacrificing the convergence. Experiments on several widely-used mathematical reasoning datasets and benchmarks with Qwen2.5-1.5B and Qwen2.5-3B show that CAPO consistently outperforms our baselines in terms of the accuracy.

## 1 Introduction

Large language models (LLMs)—from ChatGPT (Ouyang et al., 2022) to DeepSeek-R1 (Guo et al., 2025)—have made striking progress in complex reasoning tasks such as mathematical problem solving (Ahn et al., 2024; Yang et al., 2024b; Shao et al., 2024) and code generation (Jiang et al., 2024; Hou et al., 2024; Hui et al., 2024) in recent years. A key driver behind these breakthroughs is the advancement of policy optimization techniques (Sumiea et al., 2024; Schulman et al., 2015), which estimate the relative advantage of each generated response over a baseline and use it to guide parameter updates during the training process. Among them, Proximal Policy Optimization (PPO) (Schulman et al., 2017) stands out as a representative approach, using external reward and critic models to assess response quality and compute baseline scores. More recently, critic-free methods— exemplified by Group Relative Policy Optimization (GRPO) (Shao et al., 2024)—have emerged as compelling alternatives, which forgo critic models by scoring a group of responses and estimating the baseline score directly from their scores. This shift has improved both reasoning abilities and memory efficiency, sparking growing interest as a promising avenue of research (Yu et al., 2025; Liu et al., 2025; Yuan et al., 2025).

Despite their success, critic-free methods confront a fundamental challenge: *gradient conflicts* (Chen et al., 2025). Specifically, policy optimization aims to increase the likelihood of positive-advantage responses and decrease that of negative-advantage responses (Sutton et al., 1998). However, the responses within a group usually exhibit diverse reasoning paths or error patterns, resulting in gradients that point in conflicting directions. This issue intensifies during batch-wise gradient aggregation, where responses from different queries introduce additional variance. As highlighted in existing studies (Chen et al., 2025; Zhang et al., 2024; Alison et al., 2024; Chen et al., 2024), gradient conflicts can severely hinder the training performance, waste valuable computational resources, and lead to inefficient data utilization.

Unfortunately, existing methods for mitigating gradient conflicts (Liu et al., 2024; 2021; Yu et al., 2020; Sener & Koltun, 2018; Liu et al., 2023; Shi et al., 2023) fall short when applied to policy optimization techniques for LLMs. First, while these methods acknowledge that conflicts can arise from competing gradient directions, they typically treat all samples in a batch as equally important, overlooking the asymmetric roles of positive- and negative-advantage responses. Second, although some approaches use per-sample gradients to mitigate these conflicts, they often rely on computationally intensive operations (e.g., pairwise projections or matrix inversions) in the high-dimensional parameters space, rendering them impractical for LLMs. Therefore, it is highly desirable to explore conflict-mitigation methods that not only align with the nature of policy optimization, but also scale efficiently to LLMs.

To address this challenge, we propose **C**onflict-**A**ware **P**olicy **O**ptimization (**CAPO**), a novel and scalable training method that mitigates conflicts through dynamic gradient aggregation. Specifically, CAPO formulates the gradient aggregation step as a *second-order cone program (SOCP)* that seeks a direction maximally aligned with positive-advantage responses, while imposing constraints to suppress negative-advantage ones. The key intuition is that positive-advantage responses, while preferred under the reward signal, are not guaranteed to be correct or optimal; thus, encouraging them should be done with moderation. In contrast, negative-advantage responses often reflect flawed reasoning or undesirable content and should be strongly suppressed. To make the SOCP tractable for LLMs, we first apply the Lagrangian duality to substantially reduce the number of variables, and then use the Johnson-Lindenstrauss transform to project the gradients into a low-dimensional space. We further show that CAPO effectively reduces conflicts without compromising the convergence.

**Our Main Contributions.** (1) We propose CAPO, a novel and scalable policy optimization method that explicitly mitigates gradient conflicts through dynamic gradient aggregation. (2) We formulate the gradient aggregation as a second-order cone program (SOCP), taking into consideration the asymmetric roles of positive- and negative-advantage responses. We significantly accelerate its solution via the Lagrangian duality and the Johnson-Lindenstrauss transform, making it tractable and paving the way for scalable and conflict-aware optimization for LLMs. (3) We show that the dynamic gradient aggregation in CAPO effectively reduces gradient conflicts without sacrificing the convergence. (4) We conduct extensive experiments on GSM8K and MATH with Qwen2.5-1.5B and Qwen2.5-3B, showing that CAPO significantly outperforms all baselines in terms of the accuracy.

## 2 PRELIMINARIES

### 2.1 LANGUAGE MODELING AS REINFORCEMENT LEARNING

An LLM $\pi_\theta$ (with parameters $\theta$) define a conditional probability distribution over output responses $\mathbf{y} = [y_1, \ldots, y_T]$ given a query $\mathbf{x} \sim \mathcal{D}$, represented as an autoregressive policy $\pi_\theta(\mathbf{y} \mid \mathbf{x}) = \prod_{t=1}^{T} \pi_\theta(y_t \mid \mathbf{x}, \mathbf{y}_{1:t-1})$, where $\mathbf{y}_{1:0}$ is null and $\mathbf{y}_{1:t-1} = [y_1, \ldots, y_{t-1}]$ for $t = 2, \ldots, T$. To align LLMs with desired behaviors, recent work formulates language generation as a reinforcement learning (RL) problem, where the model acts as a policy that interacts with an environment by generating responses $\mathbf{y}$ to queries $\mathbf{x}$. Each response receives a reward $r(\mathbf{x}, \mathbf{y}) \in \mathbb{R}$ that reflects its quality.

Policy optimization methods aim to update the model parameters by leveraging the advantage function $A(\mathbf{x}, \mathbf{y}) = r(\mathbf{x}, \mathbf{y}) - b(\mathbf{x})$, where $b(\mathbf{x})$ is a baseline that approximates the expected reward. The overall training objective typically takes the form of an advantage-weighted log-likelihood $\max_\theta \mathbb{E}_{\mathbf{x} \sim \mathcal{D}, \mathbf{y} \sim \pi_\theta(\cdot \mid \mathbf{x})}[A(\mathbf{x}, \mathbf{y}) \log \pi_\theta(\mathbf{y} \mid \mathbf{x})]$, which encourages the model to increase the likelihood of positive-advantage responses and suppress that of negative-advantage responses.

The aforementioned formulation serves as a general template for policy optimization. Specific algorithms, such as PPO (Schulman et al., 2017), refine it by constraining the policy update to remain close to the previous policy. Instead of directly optimizing the advantage-weighted log-likelihood, PPO maximizes a clipped surrogate objective

$$\mathcal{J}_{\text{PPO}}(\theta) = \mathbb{E}_{\mathbf{x} \sim \mathcal{D}, \mathbf{y} \sim \pi_{\theta_{\text{old}}}(\cdot \mid \mathbf{x})} \left[ \min \left( \frac{\pi_\theta(\mathbf{y} \mid \mathbf{x})}{\pi_{\text{old}}(\mathbf{y} \mid \mathbf{x})} \cdot A(\mathbf{x}, \mathbf{y}), \text{clip}_{1-\varepsilon}^{1+\varepsilon} \left( \frac{\pi_\theta(\mathbf{y} \mid \mathbf{x})}{\pi_{\text{old}}(\mathbf{y} \mid \mathbf{x})} \right) \cdot A(\mathbf{x}, \mathbf{y}) \right) \right], \tag{1}$$

where $\varepsilon$ is a small hyperparameter and $\text{clip}_{\gamma_{\text{low}}}^{\gamma_{\text{high}}}(\cdot) = \text{clip}(\cdot, \gamma_{\text{low}}, \gamma_{\text{high}})$ is the clipping function.

## 2.2 CRITIC-FREE POLICY OPTIMIZATION

While PPO has become a standard method for fine-tuning LLMs, it relies heavily on external reward and critic models to compute the reward $r(\mathbf{x}, \mathbf{y})$ and baseline $b(\mathbf{x})$, respectively, introducing substantial memory and computational overhead. To address this problem, recent work has proposed critic-free methods—represented by GRPO (Shao et al., 2024), which eschew critic models by scoring a group of responses and estimating the baseline score directly from their scores. GRPO samples a group of $G$ responses $\{\mathbf{y}^{(i)}\}_{i=1}^{G}$ for each query $\mathbf{x}$ and assigns each response a scalar score $\{r^{(1)}, \ldots, r^{(G)}\}$ using a **rule-based** reward model. The advantage of each response is then computed as $A^{(i)} = \left[ r^{(i)} - \text{mean}\left(\{r^{(j)}\}_{j=1}^{G}\right) \right] \Big/ \text{std}\left(\{r^{(j)}\}_{j=1}^{G}\right)$. The overall objective of GRPO is to maximize

$$
\mathcal{J}_{\text{GRPO}}(\theta) = \mathbb{E}_{\mathbf{x}\sim\mathcal{D}, \{\mathbf{y}^{(i)}\}_{i=1}^{G}\sim\pi_{\theta_{\text{old}}}(\cdot|\mathbf{x})}
$$

$$
\left[ \frac{1}{G} \sum_{i=1}^{G} \min\left( \frac{\pi_\theta(\mathbf{y}^{(i)} \mid \mathbf{x})}{\pi_{\text{old}}(\mathbf{y}^{(i)} \mid \mathbf{x})} A^{(i)}, \text{clip}_{1-\varepsilon}^{1+\varepsilon}\left( \frac{\pi_\theta(\mathbf{y}^{(i)} \mid \mathbf{x})}{\pi_{\text{old}}(\mathbf{y}^{(i)} \mid \mathbf{x})} \right) A^{(i)} \right) - \beta \mathbb{D}_{\text{KL}}^{(i)}(\pi_\theta \| \pi_{\text{ref}}) \right], \quad (2)
$$

where $\varepsilon$ and $\beta$ are hyperparameters, and $\mathbb{D}_{\text{KL}}^{(i)}(\pi_\theta \| \pi_{\text{ref}}) = \frac{\pi_{\text{ref}}(\mathbf{y}^{(i)}|\mathbf{x})}{\pi_\theta(\mathbf{y}^{(i)}|\mathbf{x})} - \log \frac{\pi_{\text{ref}}(\mathbf{y}^{(i)}|\mathbf{x})}{\pi_\theta(\mathbf{y}^{(i)}|\mathbf{x})} - 1$.

## 2.3 GRADIENT AGGREGATION AND CONFLICTS

Abstracting away clipping and KL penalties, consider the generic objective of critic-free methods

$$
\max_\theta \ \mathbb{E}_{\mathbf{x}\sim\mathcal{D}, \{\mathbf{y}^{(i)}\}_{i=1}^{G}\sim\pi_{\theta_{\text{old}}}(\cdot|\mathbf{x})} \left[ \frac{1}{G} \sum_{i=1}^{G} \left( \frac{\pi_\theta(\mathbf{y}^{(i)} \mid \mathbf{x})}{\pi_{\text{old}}(\mathbf{y}^{(i)} \mid \mathbf{x})} A^{(i)} \log \pi_\theta(\mathbf{y}^{(i)} \mid \mathbf{x}) \right) \right]. \quad (3)
$$

According to (Lin et al., 2025), the gradient of each sample $(\mathbf{x}, \mathbf{y}^{(i)})$ can be approximated by

$$
\mathbf{g}(\mathbf{x}, \mathbf{y}^{(i)}) \approx \frac{\pi_\theta(\mathbf{y}^{(i)} \mid \mathbf{x})}{\pi_{\text{old}}(\mathbf{y}^{(i)} \mid \mathbf{x})} A^{(i)} \cdot \nabla_\theta \log \pi_\theta(\mathbf{y}^{(i)} \mid \mathbf{x}) \quad (4)
$$

and thus the aggregated gradient can by approximated by

$$
\mathbf{g}_0 \approx \frac{1}{G} \sum_{i=1}^{G} \frac{\pi_\theta(\mathbf{y}^{(i)} \mid \mathbf{x})}{\pi_{\text{old}}(\mathbf{y}^{(i)} \mid \mathbf{x})} A^{(i)} \cdot \nabla_\theta \log \pi_\theta(\mathbf{y}^{(i)} \mid \mathbf{x}). \quad (5)
$$

In this paper, **we abbreviate positive-advantage responses and negative-advantage responses as positive responses and negative responses, respectively.** Since responses with zero advantage contribute neither to the loss nor to the gradient, we exclude them from our analysis.

Despite their success, critic-free methods confront a fundamental challenge known as gradient conflicts (Chen et al., 2025). Specifically, the responses in a group usually exhibit diverse reasoning paths or error patterns, resulting in gradients that point in different directions and thus an aggregated gradient conflicting with some individual gradients. This issue intensifies during batchwise gradient aggregation, where responses from different queries introduce additional variance. As shown in Figure 1, throughout GRPO training, a non-negligible proportion of gradients from positive responses, negative responses, and all nonzero-advantage responses remain in conflict with $\mathbf{g}_0$ (i.e., their inner product with $\mathbf{g}_0$ is negative). This confirms that gradient conflicts is indeed a significant issue in critic-free policy optimization methods.

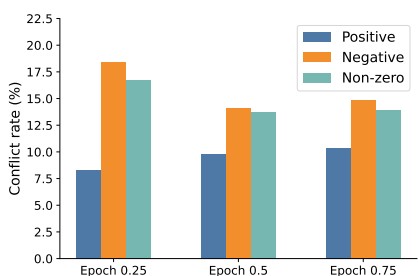

Figure 1: The proportion of gradient conflicts over training epochs. The gradients are from positive, negative, and nonzero-advantage responses.

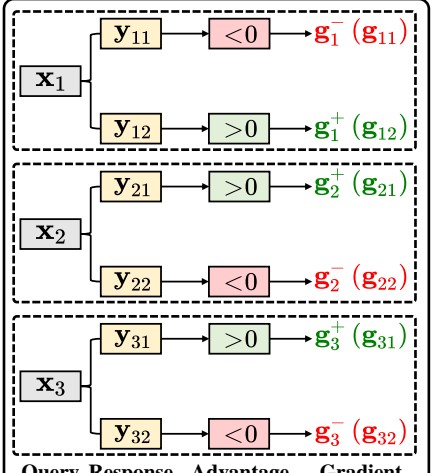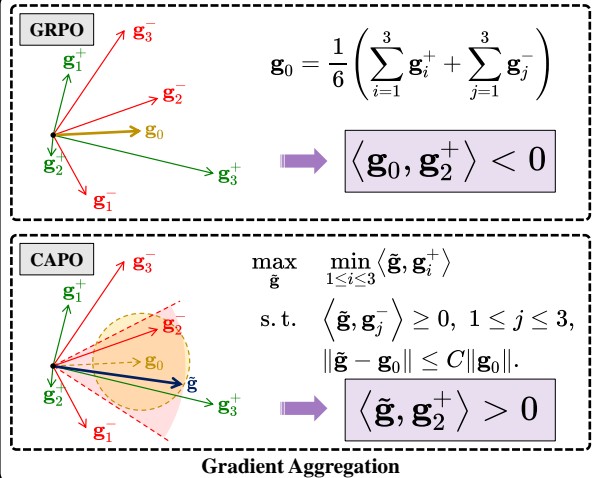

Figure 2: Comparison of the gradient aggregation strategies between CAPO and GRPO. We consider a setting with three queries $\{\mathbf{x}_i\}_{i=1}^3$, each associated with two responses $\mathbf{y}_{i1}, \mathbf{y}_{i2}$. Existing critic-free policy optimization methods (e.g., GRPO) may produce an aggregated gradient $\mathbf{g}_0$ that conflicts with some individual gradients from the responses (e.g., $\langle \mathbf{g}_0, \mathbf{g}_2^+ \rangle < 0$). In contrast, our proposed CAPO formulates the gradient aggregation as a second-order cone program (SOCP), which maximizes alignment with gradients from positive-advantage responses, while ensuring that negative-advantage responses are effectively suppressed. By solving the SOCP, we obtain a gradient $\widetilde{\mathbf{g}}$ with $\langle \widetilde{\mathbf{g}}, \mathbf{g}_2^+ \rangle > 0$.

## 3 CONFLICT-AWARE POLICY OPTIMIZATION

We propose CAPO, a novel and scalable training method that mitigates conflicts through dynamic gradient aggregation, which is shown in Figure 2. Specifically, CAPO formulates the gradient aggregation process as a second-order cone program (SOCP), taking into consideration the asymmetric roles of positive- and negative-advantage responses (Section 3.1). Leveraging the Lagrangian duality, we convert the SOCP into its dual form, which significantly reduces the number of variables (Section 3.2). To address the prohibitive computational overhead caused by the large number of parameters in LLMs, we further use the Johnson-Lindenstrauss (JL) transform to project gradients into a low-dimensional space, enabling the practical solution of the dual problem (Section 3.3). Further, we show that the dynamic gradient aggregation in CAPO effectively reduces conflicts without compromising the convergence (Section 3.4).

### 3.1 A SECOND-ORDER CONE PROGRAMMING FORMULATION FOR GRADIENT AGGREGATION

Given $m$ gradients from positive responses $\{\mathbf{g}_i^+\}_{i=1}^m$ and $n$ gradients from negative responses $\{\mathbf{g}_j^-\}_{j=1}^n$, our goal is to minimize the conflict between the aggregated gradient $\widetilde{\mathbf{g}}$ and these individual gradients. Reflecting the characteristics of rule-based reward models, we treat positive and negative responses differently:

- **For positive responses:** a positive advantage suggests that the extracted answer is correct, but it does not guarantee that the entire response is flawless—reasoning steps may still contain flaws or unnecessary content. Moreover, multiple reasoning paths can lead to the same correct answer, so instead of enforcing non-negative inner products with all positive gradients $\{\mathbf{g}_i^+\}_{i=1}^m$, we encourage $\widetilde{\mathbf{g}}$ to align with them as much as possible.

- **For negative responses:** a negative advantage indicates that the response contains a wrong answer. This implies that there is at least one critical mistake in the reasoning. Thus, we require $\widetilde{\mathbf{g}}$ to have a non-negative inner product with each gradient $\mathbf{g}_j^-$ to actively suppress these error signals.

Further, to preserve the convergence behavior of critic-free methods, we further constrain $\widetilde{\mathbf{g}}$ to lie within an $\ell_2$-ball centered around the vanilla aggregated gradient $\mathbf{g}_0$. This ensures that the final direction does not deviate too far from the vanilla optimization trajectory. With these considerations, we formulate the gradient aggregation process as a second-order cone program (SOCP):

$$\max_{\widetilde{\mathbf{g}} \in \mathbb{R}^d} \quad \min_{1 \leq i \leq m} \langle \widetilde{\mathbf{g}}, \mathbf{g}_i^+ \rangle \tag{6}$$

$$\text{s.t.} \quad \langle \widetilde{\mathbf{g}}, \mathbf{g}_j^- \rangle \geq 0, \ 1 \leq j \leq n,$$

$$\|\widetilde{\mathbf{g}} - \mathbf{g}_0\| \leq C\|\mathbf{g}_0\|,$$

where $\|\cdot\|$ is the $\ell_2$-norm and $C > 0$ is the trust region coefficient to control the size of the $\ell_2$-ball.

## 3.2 LAGRANGIAN-BASED SOLUTION TO THE SOCP

By introducing an auxiliary variable $t = \max_{1 \leq i \leq m} \left( -\langle \widetilde{\mathbf{g}}, \mathbf{g}_i^+ \rangle \right)$, the problem in Eq. (6) becomes

$$\min_{\widetilde{\mathbf{g}} \in \mathbb{R}^d, t \in \mathbb{R}} \quad t \tag{7}$$

$$\text{s.t.} \quad -\langle \widetilde{\mathbf{g}}, \mathbf{g}_i^+ \rangle - t \leq 0, \ 1 \leq i \leq m,$$

$$-\langle \widetilde{\mathbf{g}}, \mathbf{g}_j^- \rangle \leq 0, \ 1 \leq j \leq n,$$

$$\|\widetilde{\mathbf{g}} - \mathbf{g}_0\|^2 - C^2\|\mathbf{g}_0\|^2 \leq 0.$$

The Lagrangian of the problem in Eq. (7) is

$$L(\widetilde{\mathbf{g}}, t, \alpha, \beta, \lambda) = t - \sum_{i=1}^m \alpha_i \left( \langle \widetilde{\mathbf{g}}, \mathbf{g}_i^+ \rangle + t \right) - \sum_{j=1}^n \beta_j \langle \widetilde{\mathbf{g}}, \mathbf{g}_j^- \rangle + \lambda \left( \|\widetilde{\mathbf{g}} - \mathbf{g}_0\|^2 - C^2\|\mathbf{g}_0\|^2 \right), \tag{8}$$

where $\alpha_i \geq 0, \beta_j \geq 0, \lambda \geq 0$ $(1 \leq i \leq m, 1 \leq j \leq n)$ are dual variables. The first-order optimal condition implies that

$$\sum_{i=1}^m \alpha_i = 1, \quad \widetilde{\mathbf{g}} = \mathbf{g}_0 + \frac{1}{2\lambda} \left( \sum_{i=1}^m \alpha_i \mathbf{g}_i^+ + \sum_{j=1}^n \beta_j \mathbf{g}_j^- \right). \tag{9}$$

Plugging Eq. (9) into Eq. (8) leads to

$$q(\alpha, \beta, \lambda) \triangleq \inf_{\widetilde{\mathbf{g}} \in \mathbb{R}^d, t \in \mathbb{R}} L(\widetilde{\mathbf{g}}, t, \alpha, \beta, \lambda)$$

$$= -\frac{1}{4\lambda} \left\| \sum_{i=1}^m \alpha_i \mathbf{g}_i^+ + \sum_{j=1}^n \beta_j \mathbf{g}_j^- \right\|^2 - \lambda C^2 \|\mathbf{g}_0\|^2 - \left\langle \sum_{i=1}^m \alpha_i \mathbf{g}_i^+ + \sum_{j=1}^n \beta_j \mathbf{g}_j^-, \mathbf{g}_0 \right\rangle$$

$$\leq -C\|\mathbf{g}_0\| \cdot \left\| \sum_{i=1}^m \alpha_i \mathbf{g}_i^+ + \sum_{j=1}^n \beta_j \mathbf{g}_j^- \right\| - \left\langle \sum_{i=1}^m \alpha_i \mathbf{g}_i^+ + \sum_{j=1}^n \beta_j \mathbf{g}_j^-, \mathbf{g}_0 \right\rangle. \tag{10}$$

where the inequality follows from the AM–GM inequality and the equality holds if and only if $\lambda = \left\| \sum_{i=1}^m \alpha_i \mathbf{g}_i^+ + \sum_{j=1}^n \beta_j \mathbf{g}_j^- \right\| / (2C\|\mathbf{g}_0\|)$. Thus, by denoting $\xi = (\alpha_1, \ldots, \alpha_m, \beta_1, \ldots, \beta_n)^\top$ and $\mathbf{G} = [\mathbf{g}_1^+, \ldots, \mathbf{g}_m^+, \mathbf{g}_1^-, \ldots, \mathbf{g}_n^-] \in \mathbb{R}^{d \times (m+n)}$, the dual problem of the SOCP in Eq. (6) is

$$\min_{\xi \in \mathbb{R}^{m+n}} \quad C\|\mathbf{g}_0\| \cdot \|\mathbf{G}\xi\| + \langle \mathbf{G}\xi, \mathbf{g}_0 \rangle \tag{11}$$

$$\text{s.t.} \quad \sum_{i=1}^m \xi_i = 1, \quad \xi_i \geq 0 \text{ for } 1 \leq i \leq m+n.$$

## 3.3 EFFICIENT DUAL PROBLEM SOLVING FOR LLMs

Through the Lagrangian duality, we transform the original primal problem in Eq. (6) involving $d$ variables into its dual problem in Eq. (11) involving only $m + n$ variables. This significantly reduces

---

**Algorithm 1** Gradient Aggregation of CAPO

---

1: **Input:** gradients from positive responses $\{\mathbf{g}_i^+\}_{i=1}^m$ and from negative responses $\{\mathbf{g}_j^-\}_{j=1}^n$, vanilla aggregated gradient $\mathbf{g}_0$, constant $C > 0$, JL projector $P_{\mathrm{JL}} : \mathbb{R}^d \to \mathbb{R}^{\hat{d}}$

2: ▶ **Project gradients using the JL transform**

3: $\quad \hat{\mathbf{g}}_0 \leftarrow P_{\mathrm{JL}}(\mathbf{g}_0), \qquad \hat{\mathbf{g}}_i^+ \leftarrow P_{\mathrm{JL}}(\mathbf{g}_i^+)$ for $1 \le i \le m, \qquad \hat{\mathbf{g}}_j^- \leftarrow P_{\mathrm{JL}}(\mathbf{g}_j^-)$ for $1 \le j \le n$

4: ▶ **Construct the matrix and vectors for the dual problem**

5: $\quad \hat{\mathbf{G}} \leftarrow [\hat{\mathbf{g}}_1^+, \dots, \hat{\mathbf{g}}_m^+, \hat{\mathbf{g}}_1^-, \dots, \hat{\mathbf{g}}_n^-] \in \mathbb{R}^{\hat{d} \times (m+n)}$

6: $\quad \hat{\mathbf{Q}} \leftarrow \hat{\mathbf{G}}^\top \hat{\mathbf{G}} \in \mathbb{R}^{(m+n) \times (m+n)}$

7: $\quad \hat{\mathbf{q}}_0 \leftarrow \hat{\mathbf{G}}^\top \hat{\mathbf{g}}_0 \in \mathbb{R}^{m+n}$

8: ▶ **Solve the dual problem**

$$\xi^* = \underset{\xi \in \mathbb{R}^{m+n}}{\arg\min} \ C\|\mathbf{g}_0\| \left(\xi^\top \hat{\mathbf{Q}} \xi\right)^{\frac{1}{2}} + \xi^\top \hat{\mathbf{q}}_0, \quad \text{s.t.} \quad \sum_{i=1}^m \xi_i = 1, \ \xi_i \ge 0 \text{ for } 1 \le i \le m+n$$

9: ▶ **Compute the aggregated gradient**

10: $\quad \widetilde{\mathbf{g}} \leftarrow \mathbf{g}_0 + C\|\mathbf{g}_0\| \cdot (\Delta\mathbf{g}/\|\Delta\mathbf{g}\|)$, where $\Delta\mathbf{g} = \sum_{i=1}^m \xi_i^* \mathbf{g}_i^+ + \sum_{j=m+1}^{m+n} \xi_j^* \mathbf{g}_j^-$

11: **Output:** aggregated gradient $\widetilde{\mathbf{g}} \in \mathbb{R}^d$

---

the optimization complexity and computational cost, as $d$ can reach the order of billions in LLMs, while $m + n$ is bounded by the batch size and thus typically in the hundreds or thousands. However, the problem in Eq. (11) still involves a large matrix $\mathbf{G} \in \mathbb{R}^{d \times (m+n)}$, which makes the computation prohibitively expensive. In practice, even storing $\mathbf{G}$ on GPU often results in out-of-memory errors.

To make the problem tractable, we use the Johnson-Lindenstrauss (JL) transform to project the gradients into an $\hat{d}$-dimensional space, which preserves inner products within an acceptable margin of error (see Appendix C.1 for details). Formally, we denote an gradient $\mathbf{g}$ after JL projection as $\hat{\mathbf{g}} \in \mathbb{R}^{\hat{d}}$ and the resulted matrix as $\hat{\mathbf{G}} \in \mathbb{R}^{\hat{d} \times (m+n)}$. Hence, we can efficiently compute

$$\hat{\mathbf{Q}} = \hat{\mathbf{G}}^\top \hat{\mathbf{G}} \in \mathbb{R}^{(m+n) \times (m+n)}, \quad \hat{\mathbf{q}}_0 = \hat{\mathbf{G}}^\top \hat{\mathbf{g}}_0 \in \mathbb{R}^{m+n},$$

$$\|\mathbf{G}\xi\| = \left(\xi^\top \mathbf{G}^\top \mathbf{G} \xi\right)^{\frac{1}{2}} \approx \left(\xi^\top \hat{\mathbf{G}}^\top \hat{\mathbf{G}} \xi\right)^{\frac{1}{2}} = \left(\xi^\top \hat{\mathbf{Q}} \xi\right)^{\frac{1}{2}},$$

$$\langle \mathbf{G}\xi, \hat{\mathbf{g}}_0 \rangle = \xi^\top \mathbf{G}^\top \hat{\mathbf{g}}_0 \approx \xi^\top \hat{\mathbf{G}}^\top \hat{\mathbf{g}}_0 = \xi^\top \hat{\mathbf{q}}_0. \tag{12}$$

By substituting Eq. (12) into Eq. (11), we obtain a tractable problem that can be efficiently solved. In our experiments, we set $\hat{d} = 8192$ and use the CVXPY library (Diamond & Boyd, 2016) to solve the problem, which takes less than 0.1 seconds for $m + n < 128$. After obtaining $\xi^*$, the aggregated gradient is computed by $\widetilde{\mathbf{g}} = \mathbf{g}_0 + C\|\mathbf{g}_0\| \cdot (\Delta\mathbf{g}/\|\Delta\mathbf{g}\|)$, where $\Delta\mathbf{g} = \sum_{i=1}^m \xi_i^* \mathbf{g}_i^+ + \sum_{j=m+1}^{m+n} \xi_j^* \mathbf{g}_j^-$ (see Appendix C.2 for details). We summarize the overall gradient aggregation process of CAPO in Algorithm 1.

## 3.4 CONVERGENCE ANALYSIS

To address potential concerns about the compatibility between gradient conflict resolution and optimization stability, we formally establish that CAPO preserves the convergence guarantees of the baseline policy optimization method. Our analysis demonstrates that the introduced SOCP in Eq. (6)—designed to resolve gradient conflicts—do not compromise the fundamental convergence properties. Specifically, let $C^* \in [0, 1]$ denote the upper bound of the trust region coefficient, and let $C_t \in [0, C^*]$ represent its value at iteration $t$. We prove that for any $C_t \in [0, C^*]$, the gradient aggregation mechanism ensures the convergence to a neighborhood of the optimal solution, where the neighborhood size diminishes as the number of iterations $T$ increases. This guarantees that the conflict mitigation mechanism asymptotically approaches the optimal solution without fundamentally destabilizing the optimization trajectory.

Let $\mathcal{L}$ denote the average loss function and $\nabla\mathcal{L} \triangleq \mathbf{g}_0$ represent the vanilla aggregated gradient. The key insight lies in the dual role of our $\ell_2$-ball constraint $\|\widetilde{\mathbf{g}} - \mathbf{g}_0\| \le C\|\mathbf{g}_0\|$: while adaptively

resolving conflicts through the SOCP formulation, it simultaneously enforces proximity to the vanilla optimization trajectory. This geometric preservation allows us to derive Lipschitz-type bounds on the effective gradient direction, reconciling conflict mitigation with desirable convergence behavior.

**Theorem 3.1.** *Assume that the loss function $\mathcal{L}(\theta)$ is differentiable on $\mathbb{R}^d$ and its gradient $\mathbf{g}_0(\theta)$ is $H$-Lipschitz, i.e., $\|\mathbf{g}_0(\theta) - \mathbf{g}_0(\theta')\| \leq H \|\theta - \theta'\|$, where $0 < H < \infty$. Assume $\mathcal{L}^* = \inf_{\theta \in \mathbb{R}^d} \mathcal{L}(\theta) > -\infty$. With a fixed step size $\alpha$ satisfying $0 < \alpha \leq 1/H$ and $0 \leq C_t \leq C^*$ for $\forall t$, CAPO satisfies:*

1. *For $0 \leq C^* < 1$, the gap between the loss at the $T$-th iteration and the optimal loss $\mathcal{L}^*$ satisfies $\mathcal{L}(\theta_{T+1}) - \mathcal{L}^* \leq \mathcal{L}(\theta_0) - \mathcal{L}^* - \frac{\alpha}{2}(1 - C^{*2}) \sum_{t=0}^{T} \|\mathbf{g}_0(\theta_t)\|$.*

2. *When $C_t = 1$ for $\forall t$, and $0 < \alpha < 1/H$, there exists a per-iteration progress rate $\delta > 0$ such that $\mathcal{L}(\theta_T) - \mathcal{L}^* \leq \mathcal{L}(\theta_0) - \mathcal{L}^* - T\delta$.*

Under the assumption that $\mathcal{L}^* > -\infty$, as the number of iterations $T$ increases, $\mathcal{L}(\theta_T)$ asymptotically approaches $\mathcal{L}^*$. Consequently, the algorithm is guaranteed to converge to a neighborhood of $\mathcal{L}^*$. Specifically, when $T \to +\infty$, $\mathcal{L}(\theta_T) \to \mathcal{L}^*$. For detailed proof, please refer to Appendix C.3.

# 4 EXPERIMENTS

## 4.1 EXPERIMENTAL SETTINGS

**Datasets and Models.** We focus on using policy optimization to improve the mathematical reasoning ability of LLMs, which is one of the most attention-grabbing abilities of LLMs at present. We train Qwen2.5-1.5B and Qwen2.5-3B (Yang et al., 2024a) on GSM8K (Cobbe et al., 2021) and MATH (Hendrycks et al., 2021), respectively, taking into account that MATH is slightly more challenging than GSM8K. We use pretrained models instead of instruction-tuned versions to prevent them from having already seen some samples in our datasets. For more details of the datasets and models, please see Appendices B.1 and B.2.

**Baseline and Training Details.** Since the release of DeepSeek-R1 (Guo et al., 2025), there has been a surge of interest in critic-free methods, as well as various open-source efforts to replicate the DeepSeek-R1 pipeline. Among these critic-free approaches, GRPO (Shao et al., 2024) stands out as the most extensively validated method across multiple tasks, while most other methods remain unpublished or have not yet undergone peer review. For this reason, we adopt GRPO as our primary baseline, prioritizing reproducibility and empirical stability.

We build our implementation on the recently released and widely adopted veRL framework (Sheng et al., 2024), which offers an efficient and flexible RL training pipeline. We employ a rule-based reward model that evaluates both the format and correctness of responses:

$$r(\mathbf{x}, \mathbf{y}) = \begin{cases} 1.0, & \text{if } \mathbf{y} \text{ follows the correct format and correctly answers } \mathbf{x}, \\ -0.5, & \text{if } \mathbf{y} \text{ follows the correct format but incorrectly answers } \mathbf{x}, \\ -1.0, & \text{if } \mathbf{y} \text{ is incorrectly formatted.} \end{cases} \quad (13)$$

We use a learning rate of $1 \times 10^{-6}$, a prompt batch size of 64, a mini-batch size of 64, a group size of 8, a rollout temperature of 1.0, $\varepsilon = 0.2$, and $\beta = 0.001$ for CAPO and GRPO. For the JL transform, we use $\hat{d} = 8192$. We run all experiments for one epoch on 2 NVIDIA A800 GPUs (80GB) due to our limited resources. For more training details, please see Appendix B.3.

**Evaluation.** To comprehensively assess generalization, we test the models on both in-distribution and out-of-distribution (OOD) datasets. Specifically, models trained on GSM8K are evaluated on GSM8K test set, AMC 2023 (of America, 2023), and AIME 2024 (of America, 2024); while models trained on MATH are evaluated on MATH test set, AMC 2023, and AIME 2024. These datasets differ in difficulty and domain coverage, offering a rigorous evaluation setting for reasoning robustness and transfer. We employ the greedy decoding with a temperature of 0.0 and report the accuracy. For more details of the evaluation benchmarks, please see Appendix B.1.

Table 1: Performance of models trained on GSM8K and MATH with CAPO and GRPO, evaluated on the respective test set, AMC 2023, and AIME 2024.

| Dataset | Model | Method | Test Accuracy (%) | AMC 2023 (%) | AIME 2024 (%) |
|---------|-------|--------|-------------------|--------------|---------------|
| GSM8K | Qwen2.5-1.5B | GRPO | 72.33 | 15.00 | **0.00** |
| | | CAPO | **72.86** | **20.00** | **0.00** |
| MATH | Qwen2.5-3B | GRPO | 62.52 | 40.00 | 0.00 |
| | | CAPO | **62.80** | **42.50** | **3.33** |

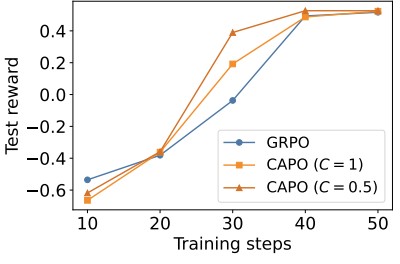

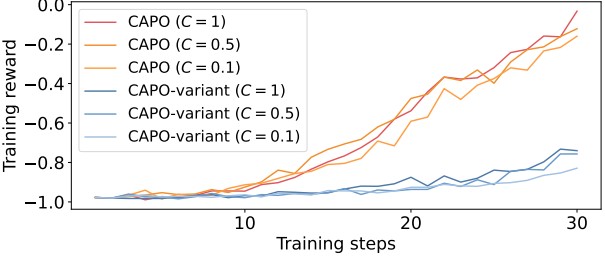

Figure 3: Test reward over training steps for GRPO and our CAPO under $C = 1$ and $C = 0.5$.

Figure 4: Training reward over training steps for CAPO and the CAPO-variant without the $\ell_2$-ball constraint, under $C = 1$, $C = 0.5$, and $C = 0.1$.

### 4.2 CAPO Improves the Mathematical Reasoning Abilities of LLMs

We present the performance of models trained on GSM8K and MATH with CAPO and GRPO, evaluated on the respective test set, AMC 2023, and AIME 2024, in Table 1. From the table, we make the following observations:

- Compared to GRPO, CAPO delivers consistently stronger performance across all benchmarks. We attribute this to the explicit resolution of gradient conflicts in CAPO, which allows for more targeted updates when positive and negative feedback signals are present simultaneously. These results suggest that even within the class of critic-free methods, incorporating structured gradient aggregation can further enhance optimization effectiveness.

- The absolute accuracy of the three methods remains low on the AIME 2024 and AMC 2023 benchmarks. This suggests that these out-of-distribution (OOD) evaluation sets pose significantly greater challenges, likely due to the increased problem complexity and reasoning depth required. One contributing factor may be the relatively narrow difficulty range of the training data—datasets such as GSM8K and MATH may not sufficiently expose the models to competition-level problem structures. In addition, the rule-based reward model, while precise in assessing final answers, provides limited supervision for intermediate reasoning steps or partial correctness, which are often crucial in solving complex questions. Despite these challenges, CAPO exhibits relatively stronger performance than GRPO on both OOD benchmarks. This suggests that the conflict-aware optimization in CAPO may help improve the robustness when generalizing to problems that differ from the training distribution, though there remains significant room for improvement overall.

### 4.3 CAPO Achieves Greater Test Reward Gains with Fewer Training Steps

We plot the test reward curves during training in Figure 3. Specifically, we show the test reward of Qwen2.5-1.5B on GSM8K, evaluated on its respective test set throughout training. We observe that CAPO yields more substantial improvements in test performance within fewer training steps compared to GRPO. This suggests that the dynamic gradient aggregation in CAPO provides more effective learning signals, enabling the model to more efficiently convert feedback into generalizable capabilities early in training. We conjecture that the explicit handling of gradient conflicts leads to more stable updates and better utilization of the available training feedback.

## 4.4 ABLATION: THE $\ell_2$-BALL CONSTRAINT IS NECESSARY

In the SOCP formulation given in Eq. (6), we introduce a constraint that bounds the new aggregated gradient within an $\ell_2$-ball centered at the vanilla aggregated gradient, i.e., $\|\widetilde{\mathbf{g}} - \mathbf{g}_0\| \leq C\|\mathbf{g}_0\|$. As discussed in Section 3.4, this constraint plays a key role in ensuring the convergence of CAPO, as it prevents the aggregated gradient from deviating too far from the vanilla gradient. To empirically verify its necessity, we conduct an ablation study on GSM8K with Qwen2.5-1.5B, comparing CAPO against a variant that replaces the $\ell_2$-ball constraint with $\|\widetilde{\mathbf{g}}\| \leq C\|\mathbf{g}_0\|$. This variant allows more freedom in selecting the gradient, without preserving proximity to $\mathbf{g}_0$. However, as shown in Figure 4, we observe a substantial drop in the speed of training reward improvement. This confirms the importance of the $\ell_2$-ball constraint in maintaining the stability of critic-free training. In addition, we observe that a larger value of $C$ generally leads to faster increases in the training reward of CAPO, although the differences in speed are not particularly significant.

## 5 RELATED WORK

**Reinforcement Learning for LLM Reasoning.** RL has become a key driver for enhancing LLM reasoning. PPO remains the dominant method (Schulman et al., 2017), though often criticized for instability and complexity (Rafailov et al., 2023; Wu et al., 2023; Yuan et al., 2023). To reduce overhead, DPO bypasses explicit reward modeling by reparameterizing preference pairs (Rafailov et al., 2023), but its performance drops under noisy or conflicting signals (Gao et al., 2024; Liang et al., 2024). This renewed interest in PPO variants led to GRPO, a critic-free approach estimating relative response quality via rule-based rewards (Shao et al., 2024), followed by widespread adoption and replication in open-source pipelines (Guo et al., 2025). Extensions include DAPO, which decouples policy and value optimization while emphasizing uncertain steps (Yu et al., 2025), and Dr.GRPO, which reduces response-length bias by simplifying normalization terms (Liu et al., 2025). These works improve training dynamics but largely overlook gradient conflicts within response groups—a key issue we address (Kim et al., 2024; Liu et al., 2024).

**Mitigating Gradient Conflicts.** Gradient conflicts have been identified as a major obstacle in machine learning, leading to inefficient learning and wasted computation (Chen et al., 2025; Zhang et al., 2024; Alison et al., 2024; Chen et al., 2024). It is essential to mitigate the gradient conflicts for improving optimization effectiveness. Most research on gradient conflict stems from multi-task learning (MTL), where gradients from different tasks may interfere with each other. This conflict is often measured via the dot product between gradient vectors (Riemer et al., 2018; Du et al., 2018). GradNorm balances gradient magnitudes across tasks to ensure uniform convergence (Chen et al., 2018). PCGrad resolves directional conflicts by projecting gradients onto orthogonal subspaces (Yu et al., 2020). MGDA seeks Pareto-optimal updates by maximizing the minimum dot product between the update and all task gradients (Sener & Koltun, 2018). ConFIG uses the pseudo-inverse to find conflict-free directions in high-dimensional spaces with theoretical Pareto guarantees (Liu et al., 2024), while CAGrad searches for updates within a local neighborhood of the average gradient under certain constraints (Liu et al., 2021). Although effective in MTL, they fall short when applied to policy optimization for LLMs due to their neglect of the asymmetric roles of positive and negative responses, as well as their computationally intensive operations. As a comparison, we propose a gradient conflict mitigation method tailored for large-model policy optimization, making a significant step toward scalable conflict-aware training for LLMs.

## 6 CONCLUSION

In this paper, we present CAPO, a novel and scalable policy optimization framework that explicitly mitigates gradient conflicts in large language models through dynamic gradient aggregation. By formulating the gradient aggregation step as a second-order cone program (SOCP), CAPO strategically amplifies the influence of positive-advantage responses while suppressing the impact of negative-advantage ones. We further enhance the scalability of our method through Lagrangian duality and the Johnson-Lindenstrauss transform, making it tractable for training large-scale models. Experiments on GSM8K and MATH using Qwen2.5-1.5B and Qwen2.5-3B demonstrate that CAPO consistently outperforms strong baselines in terms of accuracy, confirming the effectiveness of our approach.

## 7 ETHICS STATEMENT

The mathematical reasoning capabilities of LLMs hold promise for assisting mathematicians in tackling complex tasks such as theorem proving. However, they may also be used by students to complete homework assignments without engaging in independent thinking. No human subjects were directly involved. We strongly discourage any deployment outside research contexts and emphasize that reward functions and training setups are designed to encourage safe and aligned outputs. All research was conducted in accordance with the ICLR Code of Ethics, with no conflicts of interest or external influence on methodology or results.

## 8 REPRODUCIBILITY STATEMENT

To facilitate reproducibility, we provide detailed descriptions of our algorithm (CAPO) in Section 3 and Algorithm 1, including pseudo-code and key hyperparameters. Experimental setups, including data processing, reward functions, and evaluation benchmarks, are described in Section 4 and Appendix B. Where applicable, we provide references to publicly available datasets. All derivations, approximations, and additional analyses supporting the method are included in Appendix C.2. Together, these materials provide sufficient information for replication of the reported results.

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

## A  LLM Usage Statement

In preparing this manuscript, we used a large language model (LLM) in two distinct ways. First, we employed LLMs as an assistive tool for text refinement, including improving grammar, wording, and clarity. Second, LLMs themselves are the primary subject of this research: we study reinforcement learning (RL) training for LLMs. Accordingly, all experiments involve using large models for training, inference, and scoring, as part of the methodology under investigation.

All scientific content, including problem formulation, methodology, experiments, and conclusions, was developed and verified entirely by the authors. The authors take full responsibility for the integrity and accuracy of the manuscript. No LLM was credited as an author, and all substantive research contributions are attributable exclusively to the human authors.

## B  More Details about Experiments

### B.1  Datasets

We focus on enhancing the mathematical reasoning capabilities of large language models (LLMs) through policy optimization methods. Our experiments utilize two training datasets with progressive difficulty levels:

- **GSM8K** [1] (Cobbe et al., 2021): A widely-adopted arithmetic reasoning dataset containing 8,500 linguistically diverse grade-school math problems requiring multi-step reasoning. Problems are presented in natural language with solutions demonstrating explicit logical chains. We use the standard test split of 1,319 samples for evaluation.
- **MATH** [2] (Hendrycks et al., 2021): A more challenging dataset covering advanced mathematical domains including algebra, calculus, and number theory. It contains 12,500 problems with step-by-step solutions, specifically designed to evaluate deep mathematical understanding and formal reasoning capabilities. We adopt the "lighteval" subset containing 5,000 problems for efficient evaluation.

To comprehensively asses generalization, we also test the models on two challenging out-of-distribution (OOD) benchmarks AMC 2023[3] (of America, 2023) and AIME 2024[4] (of America, 2024). AMC 2023 includes high school-level multiple-choice problems designed to test creative mathematical thinking, while AIME 2024 features more advanced, open-ended questions that require multi-step reasoning and precise calculation. Both benchmarks are widely used to assess a model's ability to solve competition-style math problems.

### B.2  Models

We use the pretrained versions of Qwen2.5-1.5B and Qwen2.5-3B models. To avoid any potential contamination from pre-existing knowledge, we specifically avoid using instruction-tuned variants.

- **Qwen2.5-1.5B** [5] (Yang et al., 2024a): A compact 1.5-billion parameter model optimized for efficient training while maintaining competitive reasoning capabilities. The base model is initialized with FP32 precision and trained using LoRA (Hu et al., 2022) (rank=64, $\alpha$=128) targeting all attention matrices.
- **Qwen2.5-3B** [6] (Yang et al., 2024a): A medium-scale 3-billion parameter variant offering enhanced representational capacity. We employ identical LoRA configurations as the 1.5B model but with extended training duration to leverage its larger parameter space.

---

[1] https://huggingface.co/datasets/openai/gsm8k
[2] https://huggingface.co/datasets/DigitalLearningGmbH/MATH-lighteval
[3] https://huggingface.co/datasets/math-ai/amc23
[4] https://huggingface.co/datasets/HuggingFaceH4/aime_2024
[5] https://huggingface.co/Qwen/Qwen2.5-1.5B
[6] https://huggingface.co/Qwen/Qwen2.5-3B

### B.3 TRAINING DETAILS

We build our implementation on the recently released and widely adopted veRL framework (Sheng et al., 2024), which offers an efficient and flexible RL training pipeline. We employ a rule-based reward model that evaluates both the format and correctness of responses:

$$r(\mathbf{x}, \mathbf{y}) = \begin{cases} 1.0, & \text{if } \mathbf{y} \text{ follows the correct format and correctly answers } \mathbf{x}, \\ -0.5, & \text{if } \mathbf{y} \text{ follows the correct format but incorrectly answers } \mathbf{x}, \\ -1.0, & \text{if } \mathbf{y} \text{ is incorrectly formatted.} \end{cases} \tag{14}$$

We use a learning rate of $1 \times 10^{-6}$, a prompt batch size of 64, a mini-batch size of 64, a group size of 8, a rollout temperature of 1.0, $\varepsilon = 0.2$, and $\beta = 0.001$ for CAPO and GRPO. For the JL transform, we use $\hat{d} = 8192$. We search for the best $C$ for CAPO in $\{0.1, 0.5, 1.0\}$. We run all experiments for one epoch on 2 NVIDIA A800 GPUs (80GB).

## C MATHEMATICAL DERIVATIONS AND THEORETICAL ANALYSIS

### C.1 INTRODUCTION TO JOHNSON-LINDENSTRAUSS (JL) TRANSFORMATION

The Johnson-Lindenstrauss (JL) lemma (Johnson et al., 1984) provides theoretical guarantees for dimensionality reduction in high-dimensional spaces. Its fundamental insight demonstrates that for any finite set of points, there exists a linear projection that maps the original high-dimensional data to a low-dimensional subspace while approximately preserving both pairwise Euclidean distances and inner products within predefined error margins with high probability. This property has been widely adopted to accelerate gradient computations in large-scale optimization.

In our implementation, to reduce the computational complexity of Eq. (12), we implement a dimension reduction strategy through the JL lemma. Following the approach in (Xia et al., 2024), we project the original gradients into an 8192-dimensional space using randomized linear projections. Formally, given a gradient vector $\mathbf{g} \in \mathbb{R}^d$, we compute its compressed representation $\hat{\mathbf{g}} \in \mathbb{R}^{\hat{d}}$ through

$$\hat{\mathbf{g}} = \Pi^\top \mathbf{g}, \tag{15}$$

where $\Pi \in \mathbb{R}^{d \times \hat{d}}$ is a random projection matrix whose entries are independently drawn from a Rademacher distribution (i.e., $\pm 1$ with equal probability). Here, $d$ denotes the original gradient dimension and $\hat{d} = 8192$ specifies the reduced dimension following (Xia et al., 2024).

### C.2 DETAILED DERIVATION OF THE FINAL AGGREGATED GRADIENT $\widetilde{\mathbf{g}}$ IN CAPO

The final aggregated gradient $\widetilde{\mathbf{g}}$ in CAPO is derived through Lagrangian duality and first-order optimality conditions. We provide a step-by-step derivation below.

**Original Primal SOCP Problem** The gradient aggregation problem is formulated as a second-order cone program (SOCP) in Eq. (6):

$$\max_{\widetilde{\mathbf{g}} \in \mathbb{R}^d} \quad \min_{1 \leq i \leq m} \langle \widetilde{\mathbf{g}}, \mathbf{g}_i^+ \rangle \tag{16}$$

$$\text{s.t.} \quad \langle \widetilde{\mathbf{g}}, \mathbf{g}_j^- \rangle \geq 0, \ 1 \leq j \leq n,$$

$$\|\widetilde{\mathbf{g}} - \mathbf{g}_0\| \leq C \|\mathbf{g}_0\|,$$

where $C > 0$ controls the proximity to the original gradient $\mathbf{g}_0$.

**Step 1: Equivalent Reformulation with Auxiliary Variable** To simplify the max-min objective, we introduce an auxiliary variable $t \in \mathbb{R}$, transforming Eq. (16) into Eq. (7):

$$\min_{\widetilde{\mathbf{g}} \in \mathbb{R}^d, t \in \mathbb{R}} \quad t \tag{17}$$

$$\text{s.t.} \quad -\langle \widetilde{\mathbf{g}}, \mathbf{g}_i^+ \rangle - t \leq 0, \ 1 \leq i \leq m,$$

$$-\langle \widetilde{\mathbf{g}}, \mathbf{g}_j^- \rangle \leq 0, \ 1 \leq j \leq n,$$

$$\|\widetilde{\mathbf{g}} - \mathbf{g}_0\|^2 - C^2 \|\mathbf{g}_0\|^2 \leq 0.$$

Here, $t$ upper-bounds the negative inner products between $\widetilde{\mathbf{g}}$ and positive gradients.

**Step 2: Lagrangian Formulation**  The Lagrangian $L$ of the problem in Eq. (17) is constructed by incorporating constraints via dual variables $\alpha_i \geq 0, \beta_j \geq 0, \lambda \geq 0$:

$$L(\widetilde{\mathbf{g}}, t, \alpha, \beta, \lambda) = t - \sum_{i=1}^{m} \alpha_i \left( \langle \widetilde{\mathbf{g}}, \mathbf{g}_i^+ \rangle + t \right) - \sum_{j=1}^{n} \beta_j \langle \widetilde{\mathbf{g}}, \mathbf{g}_j^- \rangle + \lambda \left( \|\widetilde{\mathbf{g}} - \mathbf{g}_0\|^2 - C^2 \|\mathbf{g}_0\|^2 \right). \quad (18)$$

**Step 3: First-Order Optimality Conditions**  The first-order optimality conditions hold because the problem is convex and satisfies Slater's condition for $C > 0$, ensuring strong duality, while results trivially hold for $C = 0$. Taking partial derivatives of $L$ with respect to $t$ and $\widetilde{\mathbf{g}}$, and setting them to zero:

$$\frac{\partial L}{\partial t} = 1 - \sum_{i=1}^{m} \alpha_i = 0 \quad \Rightarrow \quad \sum_{i=1}^{m} \alpha_i = 1, \quad (19)$$

$$\frac{\partial L}{\partial \widetilde{\mathbf{g}}} = -\sum_{i=1}^{m} \alpha_i \mathbf{g}_i^+ - \sum_{j=1}^{n} \beta_j \mathbf{g}_j^- + 2\lambda(\widetilde{\mathbf{g}} - \mathbf{g}_0) = 0 \quad \Rightarrow \quad \widetilde{\mathbf{g}} = \mathbf{g}_0 + \frac{1}{2\lambda} \left( \sum_{i=1}^{m} \alpha_i \mathbf{g}_i^+ + \sum_{j=1}^{n} \beta_j \mathbf{g}_j^- \right). \quad (20)$$

**Step 4: Dual Problem Derivation**  Plugging Eq. (19) and Eq. (20) into Eq. (18) leads to

$$q(\alpha, \beta, \lambda) \triangleq \inf_{\widetilde{\mathbf{g}} \in \mathbb{R}^d, t \in \mathbb{R}} L(\widetilde{\mathbf{g}}, t, \alpha, \beta, \lambda)$$

$$= -\frac{1}{4\lambda} \left\| \sum_{i=1}^{m} \alpha_i \mathbf{g}_i^+ + \sum_{j=1}^{n} \beta_j \mathbf{g}_j^- \right\|^2 - \lambda C^2 \|\mathbf{g}_0\|^2 - \left\langle \sum_{i=1}^{m} \alpha_i \mathbf{g}_i^+ + \sum_{j=1}^{n} \beta_j \mathbf{g}_j^-, \mathbf{g}_0 \right\rangle. \quad (21)$$

Applying the AM–GM inequality $\frac{1}{4\lambda} \| \cdot \|^2 + \lambda C^2 \|\mathbf{g}_0\|^2 \geq C \|\mathbf{g}_0\| \| \cdot \|$, we have:

$$q(\alpha, \beta, \lambda) \leq -C\|\mathbf{g}_0\| \cdot \left\| \sum_{i=1}^{m} \alpha_i \mathbf{g}_i^+ + \sum_{j=1}^{n} \beta_j \mathbf{g}_j^- \right\| - \left\langle \sum_{i=1}^{m} \alpha_i \mathbf{g}_i^+ + \sum_{j=1}^{n} \beta_j \mathbf{g}_j^-, \mathbf{g}_0 \right\rangle, \quad (22)$$

where equality holds when:

$$\lambda = \frac{\left\| \sum_{i=1}^{m} \alpha_i \mathbf{g}_i^+ + \sum_{j=1}^{n} \beta_j \mathbf{g}_j^- \right\|}{2C\|\mathbf{g}_0\|}. \quad (23)$$

**Step 5: Final Aggregated Gradient Expression**  Let $\Delta\mathbf{g} = \sum_{i=1}^{m} \alpha_i \mathbf{g}_i^+ + \sum_{j=1}^{n} \beta_j \mathbf{g}_j^-$. From Eq. (20) and Eq. (23), substituting $\lambda$ gives:

$$\widetilde{\mathbf{g}} = \mathbf{g}_0 + \frac{\Delta\mathbf{g}}{2\lambda} = \mathbf{g}_0 + \frac{\Delta\mathbf{g}}{2 \cdot \frac{\|\Delta\mathbf{g}\|}{2C\|\mathbf{g}_0\|}} = \mathbf{g}_0 + C\|\mathbf{g}_0\| \cdot \frac{\Delta\mathbf{g}}{\|\Delta\mathbf{g}\|}. \quad (24)$$

**Step 6: Practical Implementation with JL Projection**  To compute $\Delta\mathbf{g}$ efficiently, gradients are projected into a low-dimensional space via the JL transform (Appendix C.1). Let $\hat{\mathbf{g}}_i^+ = P_{\mathrm{JL}}(\mathbf{g}_i^+)$ and $\hat{\mathbf{g}}_j^- = P_{\mathrm{JL}}(\mathbf{g}_j^-)$. The dual variables $\xi^* = (\alpha_1, \ldots, \alpha_m, \beta_1, \ldots, \beta_n)^\top$ are obtained by solving:

$$\min_{\xi} \ C\|\mathbf{g}_0\| \left( \xi^\top \hat{\mathbf{G}}^\top \hat{\mathbf{G}} \xi \right)^{1/2} + \xi^\top \hat{\mathbf{G}}^\top \hat{\mathbf{g}}_0, \quad \text{s.t. } \sum_{i=1}^{m} \xi_i = 1, \ \xi_i \geq 0, \quad (25)$$

where $\hat{\mathbf{G}} = [\hat{\mathbf{g}}_1^+, \ldots, \hat{\mathbf{g}}_m^+, \hat{\mathbf{g}}_1^-, \ldots, \hat{\mathbf{g}}_n^-]$. Substituting $\xi^*$ into $\Delta\mathbf{g}$ and Eq. (24) yields the final aggregated gradient $\widetilde{\mathbf{g}}$.

**Summary**  The derivation rigorously connects the primal SOCP problem to the dual formulation, ensuring $\widetilde{\mathbf{g}}$ balances conflict mitigation (via constraints on $\mathbf{g}_j^-$) and alignment with $\mathbf{g}_0$ (via the $\ell_2$-ball constraint). The use of JL projection maintains computational tractability without sacrificing theoretical guarantees.

## C.3 CONVERGENCE ANALYSIS

We provide the proof of CAPO's convergence in this section.

**Assumption C.1.** Assume the loss function $\mathcal{L}(\theta)$ is differentiable on $\mathbb{R}^d$ and its gradient $\nabla\mathcal{L}(\theta) \triangleq \mathbf{g}_0(\theta)$ is $H$-Lipschitz, i.e., $\|\mathbf{g}_0(x) - \mathbf{g}_0(y)\| \leq H\|x - y\|$ where $0 < H < \infty$.

**Assumption C.2.** Assume $\mathcal{L}^* = \inf_{\theta \in \mathbb{R}^d} \mathcal{L}(\theta) > -\infty$.

**Theorem C.3.** *If Assumption C.1 and Assumption C.2 hold, with a fixed step size $\alpha$ satisfying $0 < \alpha \leq 1/H$ and $0 \leq C_t \leq C^*$ for $\forall t$, the CAPO algorithm satisfies the following:*

1. *For $0 \leq C^* < 1$, the gap between the loss at the $T$-th iteration and the optimal loss $\mathcal{L}^*$ satisfies:*

$$\mathcal{L}(\theta_{T+1}) - \mathcal{L}^* \leq \mathcal{L}(\theta_0) - \mathcal{L}^* - \frac{\alpha}{2}(1 - C^{*2}) \sum_{t=0}^{T} \|\mathbf{g}_0(\theta_t)\|. \tag{26}$$

2. *When $C_t = 1$ for $\forall t$, and $0 < \alpha < 1/H$, there exists a per-iteration progress rate $\delta > 0$ such that:*

$$\mathcal{L}(\theta_T) - \mathcal{L}^* \leq \mathcal{L}(\theta_0) - \mathcal{L}^* - T\delta. \tag{27}$$

*Proof.* We will first prove Eq. (26). Consider the $t$-th optimization step and denote $\widetilde{\mathbf{g}}(\theta_t)$ as the update direction at the $t$-th iteration. Then we have:

$$
\begin{aligned}
\mathcal{L}(\theta_{t+1}) - \mathcal{L}(\theta_t) &= \mathcal{L}(\theta_t - \alpha\widetilde{\mathbf{g}}(\theta_t)) - \mathcal{L}(\theta_t) \\
&\leq -\alpha\mathbf{g}_0(\theta_t)^\top \widetilde{\mathbf{g}}(\theta_t) + \frac{H\alpha^2}{2}\|\widetilde{\mathbf{g}}(\theta_t)\|^2 \quad \text{(by the Assumption C.1)} \\
&\leq -\alpha\mathbf{g}_0(\theta_t)^\top \widetilde{\mathbf{g}}(\theta_t) + \frac{\alpha}{2}\|\widetilde{\mathbf{g}}(\theta_t)\|^2 \quad \text{(since } \alpha \leq 1/H) \\
&= -\frac{\alpha}{2}\left(\|\mathbf{g}_0(\theta_t)\|^2 + \|\widetilde{\mathbf{g}}(\theta_t)\|^2 - \|\mathbf{g}_0(\theta_t) - \widetilde{\mathbf{g}}(\theta_t)\|^2\right) + \frac{\alpha}{2}\|\widetilde{\mathbf{g}}(\theta_t)\|^2 \\
&= -\frac{\alpha}{2}\left(\|\mathbf{g}_0(\theta_t)\|^2 - \|\widetilde{\mathbf{g}}(\theta_t) - \mathbf{g}_0(\theta_t)\|^2\right) \\
&\leq -\frac{\alpha}{2}(1 - C_t^2)\|\mathbf{g}_0(\theta_t)\|^2 \quad \text{(by the constraint in Eq. (6)} \\
&\leq -\frac{\alpha}{2}(1 - C^{*2})\|\mathbf{g}_0(\theta_t)\|^2.
\end{aligned}
$$
$$\tag{28}$$
$$\tag{29}$$
$$\tag{30}$$

Using telescoping sums, we obtain the inequality

$$\mathcal{L}(\theta_{T+1}) - \mathcal{L}(\theta_0) \leq -\frac{\alpha}{2}(1 - C^{*2}) \sum_{t=0}^{T} \|\mathbf{g}_0(\theta_t)\|^2. \tag{31}$$

By introducing the $L^*$, this can be rewritten as

$$\mathcal{L}(\theta_{T+1}) - \mathcal{L}^* - (\mathcal{L}(\theta_0) - \mathcal{L}^*) \leq -\frac{\alpha}{2}(1 - C^{*2}) \sum_{t=0}^{T} \|\mathbf{g}_0(\theta_t)\|^2. \tag{32}$$

Rearranging the terms yields

$$\mathcal{L}(\theta_{T+1}) - \mathcal{L}^* \leq \mathcal{L}(\theta_0) - \mathcal{L}^* - \frac{\alpha}{2}(1 - C^{*2}) \sum_{t=0}^{T} \|\mathbf{g}_0(\theta_t)\|^2. \tag{33}$$

**Proof of equation 27.** For the case $C = 1$, we follow an analogous argument to the proof of Eq. (26). Under the step size constraint $0 < \alpha < 1/H$ (stricter than $\alpha \leq 1/H$), inequality equation 29 becomes

$$\mathcal{L}(\theta_{t+1}) - \mathcal{L}(\theta_t) < -\frac{\alpha}{2}(1 - C_t^2)\|\mathbf{g}_0(\theta_t)\|^2. \tag{34}$$

Plugging $C_t = 1 \ (\forall t)$ into Eq. 34 leads to

$$\mathcal{L}(\theta_{t+1}) - \mathcal{L}(\theta_t) < 0 \quad \forall t \geq 0. \tag{35}$$

This implies that the sequence $\{\mathcal{L}(\theta_t)\}$ is strictly monotonically decreasing. By the lower boundedness of $\mathcal{L}^*$ (Assumption C.2), there exists a constant $\delta > 0$ such that for all $t \geq 0$, we have

$$\mathcal{L}(\theta_t) - \mathcal{L}(\theta_{t+1}) \geq \delta. \tag{36}$$

Summing over $t = 0, 1, \ldots, T-1$, we obtain

$$\mathcal{L}(\theta_0) - \mathcal{L}(\theta_T) \geq T\delta. \tag{37}$$

Introducing $\mathcal{L}^*$ and rearranging the terms, we obtain

$$\mathcal{L}(\theta_T) - \mathcal{L}^* \leq \mathcal{L}(\theta_0) - \mathcal{L}^* - T\delta. \tag{38}$$

which completes the proof. $\qquad\square$

