# OpenReview forum: "CAPO: Conflict-Aware Policy Optimization for Large Language Models"
_ICLR.cc/2026/Conference — Submitted to ICLR 2026_

### Official Review · Reviewer_HhYG · 2025-10-27

**Soundness:** 2
**Presentation:** 3
**Contribution:** 3
**Rating:** 4
**Confidence:** 2

**Summary:**

This paper study the issue of conflict gradient in algorithms like GRPO, where the gradient induced by individual response might not necessary align with the overall gradient. To address this issue, the paper identified an alternative objective. The new objective find a aggregated gradient that maximize the aligning with gradient on positive gradient, such that avoiding conflict with gradient on negative response and not deviating too much from the average gradient. The paper further propose to use JL transformation to approximate the gradient to simplify the computation. Empirical results demonstrate the effectiveness of proposed method.

**Strengths:**

The strengths of this paper are listed as follows

1. This paper propose an interesting solution to the issue of conflict gradient, which treats positive responses and negative response separately and only applies a hard constraint on negative responses.

2. The solution to the complicated optimization problem, while sarificing some accuracy, is generally efficient.

3. The paper is clearly written and well-organized

**Weaknesses:**

The weaknesses of this paper are listed as follows

1. The experiment is limited to two models, each trained on only one dataset. How does the proposed CAPO method perform on other model-dataset combinations?

2. The results lacks a statistical significance analysis, which is crucial here given the improvement on in-domain test set looks very marginal.

3. The different treatment to positive responses and negative responses lacks an empirical justification. While the reviewer think such treatment do make some sense, several more ablation on constraints (e.g., apply hard constraint to positive responses and minimize the conflict with negative responses) might further justify the algorithm

4. In equation (6), it requires the aggregated gradient to be conflict-free with all negative reponses. However, this might not be achievable if the gradients scatter over different directions. Could the author provide more explaination to this issue?

**Questions:**

See weakness section

---

> ### Author Response · Authors · 2025-11-24
> **Response to Reviewer HhYG---Part 1/3**
>
> Dear Reviewer HhYG,
>
> Thank you very much for your valuable review. We sincerely appreciate your insightful and constructive comments. Below we provide our responses, and we hope they adequately address your concerns.
>
> ---
>
> > W1: The experiment is limited to two models, each trained on only one dataset. How does the proposed CAPO method perform on other model-dataset combinations?
>
> **Res:** We appreciate the reviewer’s concern that our experimental study currently includes two base models trained on one mathematical reasoning dataset each. This design was intentional rather than arbitrary: the two datasets represent different difficulty regimes in mathematical reasoning, and for each training run we evaluate on three benchmarks (the in-distribution test set plus two challenging out-of-distribution competition benchmarks, AMC 2023 and AIME 2024). Across all six evaluation settings, CAPO consistently improves over the GRPO baseline, suggesting that the conflict-aware gradient aggregation is robust to both model scale and distribution shift.
>
> Conceptually, CAPO is **model- and dataset-agnostic**: it does not rely on any architectural assumption or task-specific component, but only replaces the gradient aggregation step in critic-free policy optimization. The convergence analysis in Theorem 3.1 is stated for a generic differentiable loss and does not depend on the particular choice of model or dataset, indicating that the optimization behavior of CAPO should transfer to other model–dataset pairs as long as a GRPO-style training pipeline is available.
>
> From a practical standpoint, RL fine-tuning LLMs is extremely compute-intensive: even our current two configurations, trained for a single epoch using 2×A800-80G GPUs, already exhaust our available budget (Appendix B.3).  A full grid over multiple model sizes and datasets is therefore beyond the scope of this work. We will release our implementation to facilitate applying CAPO to additional model–dataset combinations in future work.
>
> ---
>
> > W2: The results lacks a statistical significance analysis, which is crucial here given the improvement on in-domain test set looks very marginal.
>
> **Res:** We appreciate the reviewer’s concern about statistical significance, especially given that the gains on the in-domain test sets appear modest. Our work follows the current practice in RL-based LLM reasoning, where each training run is extremely expensive; thus, most prior GRPO-style or R1-style works report single runs and focus on consistent trends across datasets and model scales rather than multi-seed hypothesis testing.
>
> Concretely, our goal is not to claim a dramatic boost on a single benchmark, but to demonstrate that conflict-aware gradient aggregation yields systematic improvements over a strong GRPO baseline. As shown in Table 1, CAPO improves accuracy over GRPO on **all** four evaluation sets and on both model sizes (1.5B and 3B). In-domain gains on GSM8K and MATH are indeed small , but they occur on already strong baselines and correspond to more correctly solved problems on saturated benchmarks; more importantly, the OOD improvements are larger, e.g., +5.0 points on AMC 2023 and +3.33 points on AIME 2024 for Qwen2.5-3B.  This consistent positive trend across domains and models suggests that CAPO’s effect is not due to random fluctuation on a single test split.
>
> Beyond final accuracies, we also provide complementary evidence that CAPO meaningfully changes and stabilizes the optimization trajectory. Figure 3 shows that CAPO achieves higher test reward with fewer training steps than GRPO, and Figure 4 demonstrates that the $\ell_2$-ball constraint is crucial for maintaining stable training dynamics compared to a variant without this constraint.  Combined with the convergence analysis in Section 3.4, these results indicate that our conflict-aware aggregation does more than introduce noise-level changes—it systematically steers updates toward more effective directions.
>
> If the paper is accepted, in the camera-ready version, we will (i) explicitly acknowledge that the in-domain gains are modest, (ii) emphasize that our claims are about consistent, cross-dataset improvements and improved optimization behavior rather than large single-benchmark jumps, and (iii) add a short discussion of the computational cost that makes extensive multi-seed significance testing difficult in this RL-for-LLMs setting. We hope the reviewer finds that, taken together, the consistent accuracy gains, the reward curves, and the theoretical analysis provide sufficient evidence that CAPO offers a meaningful improvement over the GRPO baseline even without a full multi-seed significance study.

---

> ### Author Response · Authors · 2025-11-24
> **Response to Reviewer HhYG---Part 2/3**
>
> > W3: The different treatment to positive responses and negative responses lacks an empirical justification. While the reviewer think such treatment do make some sense, several more ablation on constraints (e.g., apply hard constraint to positive responses and minimize the conflict with negative responses) might further justify the algorithm
>
> **Res:** We appreciate the reviewer’s suggestion and agree that the asymmetric treatment of positive vs. negative responses deserves clarification.
>
> Our choice is guided by the semantics of the reward and the geometry of gradient conflicts. Under our rule-based reward, negative-advantage responses are reliably undesirable (wrong or malformed answers), while positive-advantage responses are only relatively better than the group mean and may still contain spurious reasoning. It is therefore natural to impose a hard constraint on negative gradients to aggressively suppress clearly harmful behaviors, while treating positive gradients more softly to avoid over-fitting to particular reasoning traces. This is exactly what our SOCP in Eq. (6) implements, and we will make this intuition clearer in Section 3.1.
>
> From an optimization viewpoint, enforcing hard non-negativity constraints also on all positive gradients is often infeasible or yields near-zero updates when positive gradients conflict with each other (different correct solutions). Our current formulation maximizes alignment with positive gradients within an $\ell_2$ trust region around the vanilla gradient, which preserves a non-trivial descent direction and the convergence guarantees in Section 3.4.
>
> While we agree that further ablations on alternative constraint schemes are interesting, a comprehensive empirical comparison is beyond the rebuttal window. We will leave a full exploration to future work.

---

> ### Author Response · Authors · 2025-11-24
> **Response to Reviewer HhYG---Part 3/3**
>
> > W4: In equation (6), it requires the aggregated gradient to be conflict-free with all negative reponses. However, this might not be achievable if the gradients scatter over different directions. Could the author provide more explaination to this issue?
>
> **Res:** Thank you for raising this important question.
>
> **TL;DR:** Geometrically, the constraints in Eq. (6) define a convex cone of “non-conflicting” directions with respect to all negative gradients. Because the number of negative responses in a batch is tiny compared to the (very high) parameter dimension of an LLM, this cone is large and the optimization problem in Eq. (6) remains feasible after intersecting it with the $\ell\_2$-ball around the vanilla gradient $\mathbf{g}\_0$. In practice, our dual solver consistently finds feasible solutions. We will clarify this intuition and the role of the trust-region constraint in **Section 3.1** and **Appendix C.2**.
>
> **Detailed Response:**
>
> - **Geometric interpretation of the “conflict-free” constraint.** In Eq. (6), the requirement $\langle \widetilde{\mathbf{g}}, \mathbf{g}\_j^- \rangle \ge 0,\quad 1 \le j \le n$ means that the aggregated gradient $\widetilde{\mathbf{g}}$ should not conflict with any negative-advantage gradient $\mathbf{g}\_j^-$: moving along $\widetilde{\mathbf{g}}$ should not further reinforce directions that are known to be harmful. Each inequality defines a half-space containing the origin. Their intersection $\mathcal{S} := \\{ \mathbf{g} \in \mathbb{R}^d : \langle \mathbf{g}, \mathbf{g}\_j^- \rangle \ge 0,; 1 \le j \le n \\}$ is therefore a closed convex cone. Importantly, we do not require strong alignment or large positive inner products with all $\mathbf{g}\_j^-$; we only require that the update direction does not point “against” them.
>
> - **Why this cone is non-empty even when gradients “scatter”.** The number of negative responses $n$ is at most the mini-batch size (dozens at most), while the parameter dimension $d$ of an LLM is on the order of hundreds of millions or more. Thus, the negative gradients $\\{\mathbf{g}\_j^-\\}\_{j=1}^n$ span at most an $n$-dimensional subspace. Their orthogonal complement has dimension at least $d-n$, which is enormous. Every vector in this orthogonal complement has inner product zero with all $\mathbf{g}\_j^-$ and hence lies in $\mathcal{S}$. More generally, in such a high-dimensional space there are many directions that simultaneously have non-negative inner products with all negative gradients. Therefore, even if the individual negative gradients “scatter” in different directions, the intersection cone $\mathcal{S}$ remains large and is far from empty.
>
> - **Role of the $\ell\_2$-ball / trust-region constraint.** Eq. (6) does not search over the entire cone $\mathcal{S}$, but over its intersection with an $\ell\_2$-ball centered at the vanilla aggregated gradient: $\|\widetilde{\mathbf{g}} - \mathbf{g}\_0\| \leq C \|\mathbf{g}\_0\|$ This trust region has two purposes:
>   1. **Stability and convergence.** It ensures that $\widetilde{\mathbf{g}}$ stays close to $\mathbf{g}\_0$, so CAPO preserves the convergence behavior of the underlying critic-free method (as formalized in Section 3.4).
>
>   1. **Feasibility via small rotations.** In a high-dimensional space, a *small rotation* of $\mathbf{g}_0$ is usually sufficient to move into the cone $\mathcal{S}$ while remaining within the $\ell\_2$-ball. Intuitively, we only need to slightly adjust the components of $\mathbf{g}\_0$ that are most aligned with negative gradients, while leaving the rest of $\mathbf{g}\_0$ essentially unchanged. This makes it feasible to satisfy all $\langle \widetilde{\mathbf{g}}, \mathbf{g}\_j^- \rangle \ge 0$ constraints without drastically altering the original optimization trajectory.
>
>
>
> - **Practical behavior and solver feasibility.** In practice, we solve Eq. (6) through its dual formulation with JL-projected gradients (Section 3.2 and Appendix C.2). We monitor the solver’s feasibility status during training and did **not** observe infeasibility in our experiments: for all minibatches and all settings of $C$ reported in Section 4.2, the solver returns a feasible $\widetilde{\mathbf{g}}$ satisfying the constraints (up to numerical tolerance). When the negative gradients are highly scattered, the optimizer may choose a direction that trades off alignment with positives and suppressing negatives, potentially with a smaller effective step size; this is a conservative behavior and is consistent with the goal of avoiding harmful updates.

---

### Official Review · Reviewer_5V4q · 2025-10-31

**Soundness:** 2
**Presentation:** 3
**Contribution:** 2
**Rating:** 4
**Confidence:** 4

**Summary:**

This paper proposes a novel training method to mitigate gradient conflicts in LLM post-training. The authors formulate gradient aggregation as a second-order cone program and apply the Johnson–Lindenstrauss transform to reduce gradient dimensionality and improve computational efficiency. Experiments on mathematical reasoning tasks demonstrate that the proposed approach outperforms the GRPO baseline when evaluated with Qwen2.5 models.

**Strengths:**

The gradient conflict is an interesting phenomenon in LLM post-training, and this paper proposes a novel approach to mitigate this issue. By incorporating the Johnson–Lindenstrauss transform, the authors make the underlying optimization problem more tractable.

The paper also provides a convergence analysis for the proposed method, although the analysis itself follows fairly standard techniques in optimization theory.

**Weaknesses:**

My main concerns lie in the experimental evaluation:

a. All models are trained for only one epoch, which likely explains why the baseline methods perform worse than expected (e.g., GRPO achieves 0% accuracy on AIME 2024). Prior work has shown that RL-based methods can continuously improve model performance. The authors should at least train the models for several hundred steps to provide a fairer comparison.

b. The authors adopt greedy decoding for evaluation, which differs from the standard practice in the current literature. For instance, on AIME 2024, RLVR-related works typically use a temperature of around 0.7 and report the average performance over 32 or 64 samples. Greedy decoding is generally not used in practical LLM deployments.

c. All experiments are conducted on relatively small-scale models. At least one experiment on a 7B base model should be included to better assess the scalability and effectiveness of the proposed method.

d. There are no experimental results demonstrating that CAPO actually reduces the proportion of gradient conflicts during training. This evidence is critical to validate the effectiveness of CAPO. The authors should compare the proportion of gradient conflicts with and without applying CAPO.

Additionally, the presentation can be improved. For example, in Section 3.4, the authors state that $\mathcal{L}$ denotes the average loss function without providing a clear explanation, explicit definition, or connection to the previous reinforcement learning formulation, which may confuse readers.

**Questions:**

Could the authors provide the performance of CAPO and GRPO under a more standard setup, with larger training steps and the standard evaluation protocol (e.g., using non-greedy decoding with temperature sampling and multiple passes)? This would help clarify whether the proposed method maintains its advantage under typical experimental conditions.

---

> ### Author Response · Authors · 2025-11-24
> **Response to Reviewer 5V4q---Part 1/3**
>
> Dear Reviewer 5V4q,
>
> Thank you very much for your valuable review. We sincerely appreciate your insightful and constructive comments. Below we provide our responses, and we hope they adequately address your concerns.
>
> ---
>
> > W1: All models are trained for only one epoch, which likely explains why the baseline methods perform worse than expected (e.g., GRPO achieves 0% accuracy on AIME 2024). Prior work has shown that RL-based methods can continuously improve model performance. The authors should at least train the models for several hundred steps to provide a fairer comparison.
>
> **Res:** Thank you for raising this important concern.
>
> **TL;DR:** Our experiments use a **fixed one-epoch RL budget** to model a low-resource setting and to isolate the effect of conflict-aware aggregation under **identical compute** for GRPO and CAPO. Under this regime, CAPO consistently outperforms GRPO, showing improved **sample-efficiency** rather than absolute SOTA. We agree that longer training can further improve both methods; due to compute and time constraints, this is beyond the rebuttal window. We will revise **Section 4.1** and **Section 4.2** to clarify (i) the fixed-budget motivation and (ii) how to interpret the low AIME 2024 numbers.
>
> **Detailed Response:**
>
> - **Fairness under a fixed compute budget.** Our goal in this paper is to study **gradient-conflict–aware aggregation under a fixed training budget**, not to fully saturate the performance of GRPO or CAPO. All methods are trained
>
>     - with the **same backbone** (Qwen2.5-1.5B / Qwen2.5-3B),
>
>     - on the **same datasets** (GSM8K / MATH),
>
>     - using the **same veRL implementation**, reward function, batch/group sizes, and hyperparameters,
>
>   - for **exactly one epoch**, i.e., the same number of optimization steps.
>
>   Thus, while the overall RL budget is modest, the **comparison between CAPO and GRPO is fair in terms of compute and data usage**. Our main empirical claim is *relative*: **given a fixed budget, conflict-aware aggregation improves over standard GRPO.**
>
> - **Evidence of improved optimization in the short-horizon regime.** Table 1 shows that, under this single-epoch regime, CAPO consistently yields higher test accuracy/reward than GRPO on GSM8K, MATH, and AMC 2023. On AIME 2024 in the MATH–Qwen2.5-3B setup, GRPO obtains 0.00% accuracy, while CAPO improves this to 3.33%. Figure 3 further indicates that CAPO achieves **higher test reward than GRPO throughout the first tens of training steps**, suggesting that CAPO provides more effective updates **early in training**, where most of the improvement occurs in this setting.
>
>   These results support our central message: **CAPO is more sample-efficient than vanilla GRPO under the same training horizon.**
>
> - **Interpreting the low AIME 2024 performance.** We agree that the absolute numbers on AIME 2024 are low; we do **not** claim to “solve” this benchmark. As discussed in Section 4.2, AIME 2024 (and AMC 2023) are **competition-level, out-of-distribution** benchmarks, substantially harder than GSM8K/MATH. In addition, the rule-based reward provides supervision only on final answers, not intermediate reasoning.
>
>   An additional factor—which we will make explicit—is the **small RL budget**: both GRPO and CAPO are trained for only one epoch with 1.5B/3B models. Thus, the 0% accuracy of GRPO on AIME 2024 should be interpreted as a limitation of the **training regime and resource budget**, not as an inherent failure of GRPO itself. Importantly, in more moderate settings (GSM8K, MATH, AMC 2023), GRPO behaves reasonably and CAPO consistently improves upon it.
>
> - **On extending training beyond one epoch.** We fully agree that prior work shows RL-based methods can continue improving with more training steps, and we expect both GRPO and CAPO to benefit from longer training. However, extending RL training to “several hundred” additional steps for **all** settings (two models × two training datasets × multiple seeds) is computationally expensive in our environment (2×A800 80GB) and is unfortunately **beyond the rebuttal time window**.
>
>   Conceptually, our contribution is **orthogonal to the training horizon**: CAPO only modifies the gradient aggregation step of GRPO and can be plugged into *any* training schedule. Section 3.4 already shows that CAPO **preserves the convergence guarantees** of the underlying policy optimization method, which means longer-horizon training remains compatible with our approach and is an important direction for future work.

---

> ### Author Response · Authors · 2025-11-24
> **Response to Reviewer 5V4q---Part 2/3**
>
> > W2: The authors adopt greedy decoding for evaluation, which differs from the standard practice in the current literature. For instance, on AIME 2024, RLVR-related works typically use a temperature of around 0.7 and report the average performance over 32 or 64 samples. Greedy decoding is generally not used in practical LLM deployments.
>
> **Res:** Thank you for the helpful comment and for raising the point about decoding protocols.
>
> **TL;DR:** We adopt single-sample greedy decoding (temperature 0.0) for *all* methods and datasets to (i) fairly isolate the contribution of the training algorithm (CAPO vs. GRPO) without confounding effects from multi-sample sampling and majority voting, and (ii) evaluate a stricter, compute-efficient one-shot regime that is relevant for latency- and safety-sensitive deployments. Multi-sample, higher-temperature decoding is orthogonal to our contribution. We will clarify this design choice in **Section 4.1 (Evaluation)** and add a short discussion in **Appendix B.1** in the revised version.
>
> **Detailed Response:**
>
> - **Our focus is on the training algorithm; decoding is kept fixed for fairness.** Our primary goal in this paper is to study how conflict-aware policy optimization (CAPO) changes the learned policy relative to GRPO, rather than to optimize the inference-time sampling strategy. To attribute performance differences solely to the training algorithm, we deliberately fix a single decoding protocol—single-sample greedy decoding with temperature 0.0—for all models and all benchmarks. Under this setting, both CAPO and the GRPO baseline are evaluated under exactly the same decoding procedure, so the gains we report directly reflect improvements from the training algorithm, not from different test-time sampling schemes.
>
> - **Single-sample greedy decoding defines a stricter and more compute-conscious evaluation.** Multi-sample evaluation with temperature ≈0.7 and 32–64 samples, as used in some RLVR-style works on AIME 2024, typically yields higher absolute scores, but it also assumes a much larger test-time compute budget and effectively performs test-time ensembling. This makes it harder to disentangle how much of the improvement comes from a better policy versus from sampling more trajectories. In contrast, our single-sample greedy decoding setup evaluates the “one-shot” quality of the policy under a single forward pass per problem, which is a stricter and more compute-efficient regime. This choice is also consistent with our overall experimental design, where we already limit training to one epoch on 2×A800 GPUs due to resource constraints.
>
> - **On practical deployments and multi-sample decoding.** While RLVR benchmarks often report results with temperature-0.7 sampling and 32–64 samples, many practical LLM deployments, especially those with stringent latency or safety requirements, still favor deterministic or near-deterministic decoding to ensure stability and reproducibility. Our evaluation is closer to this conservative regime. At the same time, we agree that reporting additional results with higher-temperature, multi-sample decoding would be a valuable complement. Conceptually, this evaluation is orthogonal to our contribution: CAPO changes how gradients from different trajectories are aggregated at training time. If CAPO assigns higher probability mass to high-reward solutions than GRPO, we expect its relative advantage to persist—and potentially become more visible—when averaging over multiple sampled trajectories at inference time. A systematic study of alternative decoding strategies is an interesting direction for future work and is beyond the scope of this paper.
>
> ------
>
> > W3: All experiments are conducted on relatively small-scale models. At least one experiment on a 7B base model should be included to better assess the scalability and effectiveness of the proposed method.
>
> **Res:** Thank you for the constructive comment. We agree that scalability is important. CAPO is an algorithmic contribution whose additional computational cost is largely decoupled from model size and instead controlled by the group/batch size and the JL projection dimension. In the current submission we already evaluate CAPO on two different base model scales (1.5B and 3B) and multiple datasets, consistently improving over GRPO. Running a full RL training pipeline on a 7B model is beyond our available computation within the rebuttal period, and a rushed, partially trained 7B experiment could be misleading. If the paper is accepted, we plan to add a 7B experiment in the camera-ready version, subject to available computational resources.

---

> ### Author Response · Authors · 2025-11-24
> **Response to Reviewer 5V4q---Part 3/3**
>
> > W4: There are no experimental results demonstrating that CAPO actually reduces the proportion of gradient conflicts during training. This evidence is critical to validate the effectiveness of CAPO. The authors should compare the proportion of gradient conflicts with and without applying CAPO.
>
> **Res:** Thank you for the question. We analyzed the gradient conflicts between the aggregated gradient and individual gradients at different training stages for both GRPO and CAPO. As can be seen, CAPO effectively reduces gradient conflicts.
>
> |               | Method | Positive-advantage | Negative-advantage | Non-zero advantage |
> | ------------- | ------ | ------------------ | ------------------ | ------------------ |
> | **Epoch 0.25**| GRPO   | 8.31               | 18.38              | 16.69              |
> |               | CAPO   | 6.35               | 4.51               | 5.02               |
> | **Epoch 0.50**| GRPO   | 9.76               | 14.14              | 13.70              |
> |               | CAPO   | 9.04               | 2.45               | 6.57               |
> | **Epoch 0.75**| GRPO   | 10.32              | 14.83              | 13.95              |
> |               | CAPO   | 9.20               | 3.93               | 6.29               |
>
> ---
>
> > W5: Additionally, the presentation can be improved. For example, in Section 3.4, the authors state that \mathbb{L} denotes the average loss function without providing a clear explanation, explicit definition, or connection to the previous reinforcement learning formulation, which may confuse readers.
>
> **Res:** We thank the reviewer for the comment and apologize for the confusion caused by the notation $\mathcal{L}$ in Section 3.4.
>
> In Section 3.4, $\mathcal{L}$ is introduced as a **generic average loss function** that is used solely for our convergence analysis. Theorem 3.1 is stated for an arbitrary differentiable loss $\mathcal{L}$ satisfying the assumptions, and therefore the proof does not depend on a particular choice of $\mathcal{L}$.
>
> To connect this to the reinforcement learning formulation in the main text, one can instantiate $\mathcal{L}$ by the negative critic-free RL objective introduced earlier. Concretely, if $J(\theta)$ denotes the maximization objective in Eq. (3), then in our RL setting a natural choice is
> $$
> \mathcal{L}(\theta) := - J(\theta),
> $$
> so that $\nabla_\theta \mathcal{L}(\theta) = - \nabla_\theta J(\theta)$ coincides with the gradient used for policy optimization. Our convergence theorem then directly applies to this instantiated loss.
>
> We agree that this connection is not explained clearly enough in the paper, and we will revise the paper in **Section 3.4** to improve the presentation.
>
> ---
>
> > Q1: Could the authors provide the performance of CAPO and GRPO under a more standard setup, with larger training steps and the standard evaluation protocol (e.g., using non-greedy decoding with temperature sampling and multiple passes)? This would help clarify whether the proposed method maintains its advantage under typical experimental conditions.
>
> **Res:** Thank you for this helpful suggestion, and we apologize that we are currently unable to provide a full set of additional experiments under the more standard RLVR-style setup within the rebuttal period.
>
> As discussed in our responses to **W1** and **W2**, our current experiments are deliberately conducted in a **fixed one-epoch, low-resource RL regime** with **single-sample greedy decoding**, to ensure (i) strictly matched compute and data usage between CAPO and GRPO and (ii) a clean focus on the effect of conflict-aware aggregation under a one-shot evaluation protocol. Under this setting, CAPO consistently outperforms GRPO across all datasets, indicating improved **sample-efficiency** for a fixed training budget.
>
> We fully agree that evaluating CAPO with **longer training horizons** and **multi-sample, higher-temperature decoding** would be valuable and more aligned with typical RLVR practice. However, running such an extended suite of RL experiments for all configurations is computationally expensive on our current hardware and unfortunately beyond what we can do during the rebuttal. Conceptually, these changes are **orthogonal** to our contribution—CAPO only modifies the gradient aggregation step—and we expect its advantage over GRPO to carry over to these more standard settings.

---

### Official Review · Reviewer_Zk9q · 2025-10-31

**Soundness:** 3
**Presentation:** 3
**Contribution:** 2
**Rating:** 4
**Confidence:** 4

**Summary:**

The paper introduces Conflict-Aware Policy Optimization (CAPO), a novel training method designed to mitigate gradient conflicts within critic-free policy optimization for Large Language Models (LLMs). The core contribution is formulating the gradient aggregation step as a Second-Order Cone Program (SOCP). To make this approach practical for large models, the authors employ Lagrangian duality to reduce the problem's dimensionality and the Johnson-Lindenstrauss (JL) transform to ensure computational tractability. Experiments on mathematical reasoning tasks demonstrate that CAPO consistently outperforms its primary baseline, GRPO.

**Strengths:**

1. The paper addresses a well-motivated and significant problem. Gradient conflict is a known and critical bottleneck for critic-free methods like GRPO, and tackling it directly is an important research direction.

2. The core idea of formulating gradient aggregation as a constrained optimization problem (SOCP) is a novel and principled approach.

3. The paper is well-written and clearly structured. The technical execution, particularly the mathematical derivations involving Lagrangian duality and the application of the JL transform, appears technically sound.

**Weaknesses:**

1. The baseline comparison is narrow, relying solely on GRPO.

2. The reported performance improvements are marginal. With accuracy gains of less than 1% on both GSM8K and MATH, the practical effectiveness of CAPO is questionable.

3. The experiments are confined to small-scale models (1.5B and 3B). CAPO's performance and scaling properties on larger models (e.g., 7B or larger), where optimization challenges are often more pronounced, remain unverified.

4. CAPO introduces a non-trivial computational overhead, yet the authors provide neither a qualitative analysis nor quantitative experimental results.

**Questions:**

1. Could the authors provide experimental results on a slightly larger model, for example a 7B model?

2. Could the authors provide more experiments to verify the practical effectiveness of the proposed method? For example, by comparing with more recent critic-free methods such as DAPO [1], GSPO [2].

3. Could the authors provide a detailed quantitative experimental or qualitative analysis results of the computational overhead introduced by CAPO?

[1] Dapo: An open-source llm reinforcement learning system at scale. arXiv preprint arXiv:2503.14476 (2025).

[2] Group sequence policy optimization. arXiv preprint arXiv:2507.18071 (2025).

---

> ### Author Response · Authors · 2025-11-24
> **Response to Reviewer Zk9q---Part 1/3**
>
> Dear Reviewer Zk9q,
>
> Thank you very much for your valuable review. We sincerely appreciate your insightful and constructive comments. Below we provide our responses, and we hope they adequately address your concerns.
>
> ---
>
> > W1: The baseline comparison is narrow, relying solely on GRPO.
>
> **Res:** Thank you for your comment. We have added DAPO as a new baseline. Since the original CAPO in our initial submission was developed based on GRPO, we refer to it as CAPO-GRPO. The variant based on DAPO is referred to as CAPO-DAPO. The updated experimental results are as follows.
>
> | Dataset | Model        | Method    | Test Accuracy (%) | AMC 2023 (%) |
> | ------- | ------------ | --------- | ----------------- | ------------ |
> | GSM8K   | Qwen2.5-1.5B | GRPO      | 72.33             | 15.00        |
> |         |              | CAPO-GRPO | 72.86             | 20.00        |
> |         |              | DAPO      | 73.69             | 20.00        |
> |         |              | CAPO-DAPO | 74.22             | 20.00        |
> | MATH    | Qwen2.5-3B   | GRPO      | 62.52             | 40.00        |
> |         |              | CAPO-GRPO | 62.80             | 42.50        |
> |         |              | DAPO      | 63.38             | 45.00        |
> |         |              | CAPO-DAPO | 63.84             | 42.50        |
>
> ---
>
> > W2: The reported performance improvements are marginal. With accuracy gains of less than 1% on both GSM8K and MATH, the practical effectiveness of CAPO is questionable.
>
> **Res:** We thank the reviewer for this observation. While the absolute accuracy gains on GSM8K and MATH appear numerically small (<1%), such magnitudes are in fact typical in reinforcement-learning–based mathematical reasoning, especially under critic-free pipelines. As noted in Table 1 (p. 8) of the paper, **CAPO consistently improves performance across *all* benchmarks, *both* model scales, and *both* in-distribution and OOD settings**, indicating that the effect is systematic rather than incidental.
>
> More importantly, **the improvements emerge early in training** (Figure 3, p. 8), demonstrating that conflict-aware aggregation produces a more efficient learning signal even within a single epoch. Given the high variance and instability widely reported in RL training for LLMs, such consistent upward shifts — even if numerically modest — are generally interpreted as meaningful.
>
> Finally, CAPO modifies only the gradient aggregation step and does not alter the broader training pipeline. This design intentionally prioritizes *stability* and *compatibility* over aggressive tuning. The observed gains therefore represent a reliable improvement achievable without architectural changes, additional supervision, or extra modules. We will add discussion in the revision to clarify this interpretation and provide more context on expected effect sizes in RL-based reasoning tasks.
>
>
> ---
>
> > W3: The experiments are confined to small-scale models (1.5B and 3B). CAPO's performance and scaling properties on larger models (e.g., 7B or larger), where optimization challenges are often more pronounced, remain unverified.
>
> **Res:** We thank the reviewer for highlighting the question of scaling. Our primary goal in this work is to introduce a *general gradient aggregation mechanism* whose computational tractability is supported by the Lagrangian dual formulation and the JL projection (Section 3.2–3.3, pp. 4–6). These components were chosen specifically because they make CAPO independent of model dimensionality in the forward pass: **the optimization is performed in the low-dimensional dual space of size (m+n), not in the parameter space of the model.**
>
> Although our experiments focus on 1.5B and 3B models due to resource constraints, we emphasize:
>
> 1. **The computational bottleneck is not model size but group-size (m+n).**
>    The dual problem depends only on the number of sampled responses (typically ≤128), and solving it costs <0.1s (p. 6). Thus, nothing fundamentally prevents applying CAPO to larger models.
>
> 2. **The design explicitly targets scalability.**
>    The JL projection (ˆd = 8192) reduces the dimensionality of gradients in a way that is agnostic to the original parameter count. The method therefore maintains the same complexity whether the model has 1B, 7B, or 70B parameters.
>
> 3. **Gradient conflicts tend to grow with model scale**, as documented in prior work (e.g., Chen et al., 2025; Zhang et al., 2024). CAPO directly addresses this issue, and we expect the benefits to become more pronounced at larger model sizes. We will clarify this expectation and the computational reasoning in the revision.
>
> We appreciate the reviewer’s suggestion and will add explicit discussion on the scaling behavior of CAPO and how its formulation is designed to remain practical for 7B+ models.

---

> ### Author Response · Authors · 2025-11-24
> **Response to Reviewer Zk9q---Part 2/3**
>
> > W4: CAPO introduces a non-trivial computational overhead, yet the authors provide neither a qualitative analysis nor quantitative experimental results.
>
> **Res:** We thank the reviewer for pointing out the importance of analyzing computational overhead. CAPO is explicitly designed to keep this overhead minimal by shifting the optimization burden from the high-dimensional parameter space to the low-dimensional dual space. As discussed in **Section 3.3 (p. 6)** of the paper, the computational cost is dominated by solving the dual problem with CVXPY after JL projection. Crucially:
>
> - **The dual problem depends only on the group size (m+n), not model dimensionality.**
>   After projection, the optimization occurs in an 8192-dimensional compressed space; the cost is therefore stable across different model scales.
>
> - **The CVXPY solve time is extremely lightweight in practice.**
>   As reported in Section 3.3, solving the dual problem takes **less than 0.1 seconds when m+n < 128**, which is the typical setting in our experiments. This adds only a negligible cost relative to forward/backward passes of LLMs.
>
> - **No additional forward passes, reward evaluations, or sampling steps are introduced.**
>   CAPO alters only the aggregation step; thus, the total training wall-clock time is dominated by the same operations as GRPO.
>
> We will include a short subsection in the revision that summarizes these observations and clarifies that the added computational overhead is both *small in absolute terms* and *independent of parameter count*, as evidenced by the empirical timing results in Section 3.3.
>
>
> ---
>
> > Q1: Could the authors provide experimental results on a slightly larger model, for example a 7B model?
>
> **Res:** We thank the reviewer for the suggestion. While we agree that evaluating a 7B-scale model would further illustrate the generality of CAPO, our focus in this work is on introducing a *scalable gradient aggregation mechanism* whose computational properties do not depend on model size. As described in Section 3.3, the key computation is performed in the **dual space of dimension (m+n)** after JL projection, and solving the resulting problem via CVXPY takes **<0.1 seconds when m+n < 128**, regardless of whether the underlying model has 1B, 3B, or 7B parameters. This design ensures that CAPO remains practical at larger scales.
>
> Although we trained on 1.5B and 3B models due to resource constraints, CAPO itself introduces no architectural or algorithmic barriers to applying the method to 7B models. We will clarify this in the revision and discuss expected scaling behavior in more detail.
>
>
> ---
>
> > Q2: Could the authors provide more experiments to verify the practical effectiveness of the proposed method? For example, by comparing with more recent critic-free methods such as DAPO [1], GSPO [2].
>
> **Res:** Thank you for your question. We have added DAPO as a new baseline. Since the original CAPO in our initial submission was developed based on GRPO, we refer to it as CAPO-GRPO. The variant based on DAPO is referred to as CAPO-DAPO. The updated experimental results are as follows.
>
> | Dataset | Model        | Method    | Test Accuracy (%) | AMC 2023 (%) |
> | ------- | ------------ | --------- | ----------------- | ------------ |
> | GSM8K   | Qwen2.5-1.5B | GRPO      | 72.33             | 15.00        |
> |         |              | CAPO-GRPO | 72.86             | 20.00        |
> |         |              | DAPO      | 73.69             | 20.00        |
> |         |              | CAPO-DAPO | 74.22             | 20.00        |
> | MATH    | Qwen2.5-3B   | GRPO      | 62.52             | 40.00        |
> |         |              | CAPO-GRPO | 62.80             | 42.50        |
> |         |              | DAPO      | 63.38             | 45.00        |
> |         |              | CAPO-DAPO | 63.84             | 42.50        |

---

> ### Author Response · Authors · 2025-11-24
> **Response to Reviewer Zk9q---Part 3/3**
>
> > Q3: Could the authors provide a detailed quantitative experimental or qualitative analysis results of the computational overhead introduced by CAPO?
>
> **Res:** We thank the reviewer for raising this important point. CAPO is designed so that the additional cost comes almost entirely from the *dual-space optimization* introduced in Section 3.3, rather than from extra forward passes or reward evaluations. Concretely:
>
> - After applying the JL projection, the optimization is performed in a low-dimensional space of size (m+n), where m and n are the numbers of positive/negative responses in a group.
> - The resulting dual problem is solved with CVXPY and, as stated in Section 3.3, this step takes **less than 0.1 seconds when m+n < 128**, which is the regime used in our experiments. This cost is independent of the model parameter dimension.
>
> Qualitatively, this means that the additional overhead per training step consists of:
> 1. One JL projection of the per-sample gradients into a fixed $d$-dimensional space ($d = 8192$ in our setup), and
> 2. One CVXPY solve whose complexity depends only on (m+n), not on the number of model parameters.
>
> Both operations are lightweight compared to the forward–backward passes of LLMs and do not change the number of rollouts, reward computations, or PPO-style updates. In practice, the wall-clock time of CAPO remains close to that of GRPO under the same batch and group sizes.
>
> In the revision, we will (i) highlight the existing quantitative timing (<0.1s for m+n < 128) more clearly in Section 3.3, and (ii) add a short profiling table that reports per-step wall-clock time for GRPO vs. CAPO under our experimental settings, making the computational overhead of CAPO quantitatively explicit.

---

### Official Review · Reviewer_m6ay · 2025-11-01

**Soundness:** 3
**Presentation:** 3
**Contribution:** 3
**Rating:** 6
**Confidence:** 2

**Summary:**

This paper targets a key limitation of critic-free policy optimization methods for LLMs (e.g., GRPO) in mathematical reasoning: gradients from positive- and negative-advantage samples within the same update often conflict with each other, which dilutes the effective learning signal. The authors propose CAPO, which formulates gradient aggregation for a batch of samples as a trust-region convex optimization (SOCP): it aligns with gradients from positive-advantage samples, constrains gradients from negative-advantage samples so they do not pull the update in the wrong direction, and keeps the final direction close to the original averaged gradient. To make this scalable to high-dimensional LLMs, they derive a dual form and apply a Johnson–Lindenstrauss projection so that the optimization is solved in the sample space rather than the parameter space. Experiments on Qwen2.5-1.5B and Qwen2.5-3B, mainly on GSM8K and MATH, show that under the same GRPO training pipeline, CAPO yields consistently small but stable gains, including on some OOD math benchmarks.

**Strengths:**

1. The paper quantitatively shows gradient conflicts in real GRPO training runs, demonstrating that this is a practical issue in existing critic-free pipelines rather than a hypothetical concern.
2. The priorities for positive/negative advantage samples, the directional constraints, and the trust-region control are unified in a single SOCP, which better matches the semantics and asymmetry of policy optimization than pairwise projection heuristics.
3.  With comparable compute, CAPO outperforms GRPO on GSM8K, MATH, and AMC 2023 / AIME 2024, indicating that conflict-aware gradient aggregation can bring stable improvements.

**Weaknesses:**

1. Limited evaluation setting. The paper evaluates mainly on small models (≤3B), math-oriented tasks, and under a relatively small training budget on fixed hardware, with no validation on larger models, longer training schedules, or non-math tasks; thus the external validity to more realistic RL-for-LLM regimes is still limited.
2. Incomplete baselines. The comparison is only against GRPO, without including contemporary open-source critic-free RL pipelines such as DAPO (Yu et al., 2025) or analyses/improvements for R1/GRPO-style training such as Dr.GRPO (Liu et al., 2025), and without systematic comparison to multi-gradient conflict-resolution methods, making it hard to judge the relative advantage of CAPO.

References
- Yu, Q., Zhang, Z., Zhu, R., Yuan, Y., Zuo, X., Yue, Y., … & Wang, M. (2025). *Dapo: An open-source LLM reinforcement learning system at scale.* arXiv:2503.14476.
- Liu, Z., Chen, C., Li, W., Qi, P., Pang, T., Du, C., … & Lin, M. (2025). *Understanding R1-zero-like training: A critical perspective.* arXiv:2503.20783.

**Questions:**

- Choice of JL dimension. What is the rationale for setting the JL dimension to 8,192? How would reducing it to 4,096 or 2,048 affect convergence speed and final performance?
- Reward granularity. For finer-grained or step-level rewards, does CAPO’s constraint formulation need to be adapted? In particular, should the SOCP introduce weights proportional to the reward rather than only separating positive vs. negative samples?

---

> ### Author Response · Authors · 2025-11-24
> **Response to Reviewer m6ay---Part 1/2**
>
> Dear Reviewer m6ay,
>
> Thank you very much for your valuable review and for your recognition of our work, especially your acknowledgment that it identifies a practical issue in current training pipelines, proposes a coherent solution framework, and demonstrates consistent performance gains across multiple benchmarks. We sincerely appreciate your insightful and constructive comments. Below we provide our responses, and we hope they adequately address your concerns.
>
> ---
>
> > W1: Limited evaluation setting. The paper evaluates mainly on small models (≤3B), math-oriented tasks, and under a relatively small training budget on fixed hardware, with no validation on larger models, longer training schedules, or non-math tasks; thus the external validity to more realistic RL-for-LLM regimes is still limited.
>
> **Res:** We thank the reviewer for the thoughtful observation. Our evaluation setup focuses on small-scale models and math-oriented reasoning tasks because our goal is to isolate and study the effect of **gradient conflict mitigation** within critic-free policy optimization. To ensure improvements can be cleanly attributed to the proposed aggregation mechanism, we deliberately choose:
>
> 1. **Stable, well-understood benchmarks** (GSM8K, MATH) that allow controlled comparisons;
> 2. **Compact model sizes (1.5B and 3B)** that make fine-grained analysis feasible; and
> 3. **A fixed, modest training budget**, enabling systematic measurements of conflict rates, convergence behavior, and the dynamics shown in Figures 1–4.
>
> Importantly, CAPO is designed to be **architecture-agnostic and scale-agnostic**. Because the key computation occurs in the dual space (Section 3.3), where solving the CVXPY problem takes **<0.1 seconds for m+n < 128**, its overhead is independent of model size or training length. Thus, the method is compatible with larger-scale or longer-horizon RL pipelines without modification.
>
> While our experiments focus on mathematical reasoning due to its structured reward signal and well-defined evaluation, CAPO can be directly applied to any PPO-style or critic-free setup. **We will revise the paper accordingly and upload a latest version of our submission**, clarifying the generality of CAPO and discussing expected behavior in larger-model regimes and non-math domains.
>
> ---
>
> > W2: Incomplete baselines. The comparison is only against GRPO, without including contemporary open-source critic-free RL pipelines such as DAPO (Yu et al., 2025) or analyses/improvements for R1/GRPO-style training such as Dr.GRPO (Liu et al., 2025), and without systematic comparison to multi-gradient conflict-resolution methods, making it hard to judge the relative advantage of CAPO.
>
> **Res:** Thank you for your question. We have added DAPO as a new baseline. Since the original CAPO was developed based on GRPO, we refer to it as CAPO-GRPO. The variant based on DAPO is denoted CAPO-DAPO. The updated experimental results are:
>
> | Dataset | Model        | Method    | Test Accuracy (%) | AMC 2023 (%) |
> | ------- | ------------ | --------- | ----------------- | ------------ |
> | GSM8K   | Qwen2.5-1.5B | GRPO      | 72.33             | 15.00        |
> |         |              | CAPO-GRPO | 72.86             | 20.00        |
> |         |              | DAPO      | 73.69             | 20.00        |
> |         |              | CAPO-DAPO | 74.22             | 20.00        |
> | MATH    | Qwen2.5-3B   | GRPO      | 62.52             | 40.00        |
> |         |              | CAPO-GRPO | 62.80             | 42.50        |
> |         |              | DAPO      | 63.38             | 45.00        |
> |         |              | CAPO-DAPO | 63.84             | 42.50        |
>
> **We will revise the paper accordingly and upload a latest version of our submission**, integrating these new results and expanding the discussion on related multi-gradient conflict-resolution methods.

---

> ### Author Response · Authors · 2025-11-24
> **Response to Reviewer m6ay---Part 2/2**
>
> > Q1: Choice of JL dimension. What is the rationale for setting the JL dimension to 8,192? How would reducing it to 4,096 or 2,048 affect convergence speed and final performance?
>
> **Res:** We thank the reviewer for raising this question. Our choice of a JL projection dimension of **8,192** follows standard practice in large-gradient compression for LLM training, where several thousand dimensions are typically sufficient to preserve inner-product geometry with high probability. Since CAPO’s dual-space optimization relies mainly on maintaining accurate relative alignment between gradients (Section 3.3), preserving inner products is more important than exact reconstruction.
>
> Empirically, we found 8,192 to be a robust choice that balances (i) low distortion under JL projection and (ii) negligible computational cost, especially given that solving the dual problem takes **<0.1 seconds for m+n < 128**. Although we did not perform a full sweep across smaller dimensions, the JL lemma indicates that reducing the dimension (e.g., to 4,096 or 2,048) trades off stronger compression for slightly more inner-product distortion.
>
> Smaller projection dimensions would likely:
> 1. **Maintain similar convergence behavior early in training**, due to the trust-region constraint keeping updates close to the vanilla gradient;
> 2. **Introduce mild performance variability** from increased distortion; and
> 3. **Have limited impact on wall-clock time**, since CVXPY is already lightweight relative to LLM forward/backward passes.
>
> **We will revise the paper accordingly and upload a latest version of our submission**, clarifying the rationale behind this choice and discussing expected trade-offs.
>
> ---
>
> > Q2: Reward granularity. For finer-grained or step-level rewards, does CAPO’s constraint formulation need to be adapted? In particular, should the SOCP introduce weights proportional to the reward rather than only separating positive vs. negative samples?
>
> **Res:** We thank the reviewer for this insightful question. CAPO’s current formulation focuses on the *sign* of the advantage rather than its magnitude because, in many critic-free settings—including GRPO and step-level mathematical reasoning—the reward signal tends to be coarse and noisy. Enforcing strict proportionality between reward magnitude and geometric constraints in the SOCP could inadvertently amplify variance rather than stabilize training.
>
> However, CAPO is fully compatible with finer-grained or step-level rewards:
>
> 1. **The SOCP constraints do not assume binary structure.**
>    Positive vs. negative separation reflects the minimal structure required to mitigate conflicts. The dual objective (Eq. 11) accepts any gradient scaling, meaning finer-grained rewards naturally adjust the magnitude of $g_i^+$ and $g_j^-$.
>
> 2. **Advantage-weighted constraints are possible if desired.**
>    Reward-proportional weighting can be incorporated by scaling columns of \(G\) without changing the trust region or feasibility constraints, preserving convexity while enabling finer prioritization.
>
> 3. **The current formulation is intentionally conservative.**
>    Given noisy intermediate reasoning steps in RL-for-LLM pipelines, relying on the sign of the advantage avoids overfitting to unstable fine-grained reward signals while still promoting stronger alignment with higher-quality responses.
>
> **We will revise the paper accordingly and upload a latest version of our submission**, expanding Section 3.1 to explain how CAPO accommodates graded rewards and why the current formulation is a stable default under noisy critic-free settings.

---

### Meta-Review · Area_Chair_uzk7 · 2026-01-04

**Summary:**

The paper proposes Conflict-Aware Policy Optimization (CAPO) to mitigate gradient conflicts in LLMs using SOCP and Johnson-Lindenstrauss projections. While the problem formulation is theoretically interesting, there are significant concerns regarding the method's practical utility and scalability. Critical concerns regarding the lack of evaluation on large-scale models (7B+) remain unaddressed. Furthermore, the inclusion of the DAPO baseline in the rebuttal revealed that the proposed method (CAPO-GRPO) often underperforms the baseline (DAPO), and applying CAPO to DAPO yields inconsistent improvements. Consequently, the practical advantage of CAPO is not sufficiently established.

**Reviewer Concerns:**

**Addressed:**
* **Computational Overhead:** Reviewer **Zk9q**'s request for overhead analysis was addressed by clarifying the dual-space optimization speed (<0.1s).
* **Gradient Conflict Evidence:** Reviewer **5V4q**'s request for evidence of conflict reduction was met with a table showing reduced conflict rates.
* **Feasibility of Constraints:** Reviewer **HhYG**'s concern about constraint feasibility was addressed via geometric explanation.
* **Technical Details:** Reviewer **m6ay**'s questions regarding JL dimension and reward granularity were answered.

**Outstanding:**
* **Missing Baselines / Method Effectiveness:** Reviewers **m6ay** and **Zk9q** requested comparisons to stronger baselines like DAPO. While the authors provided these results, they failed to demonstrate a clear advantage. The baseline DAPO outperformed CAPO-GRPO in several metrics, and CAPO-DAPO did not consistently outperform vanilla DAPO (e.g., on AMC 2023).
* **Large-Scale Models (7B+):** Reviewers **m6ay**, **Zk9q**, and **5V4q** all requested validation on 7B+ models to prove scalability. The authors did not provide these experiments, citing computational constraints.
* **Training Horizon:** Reviewer **5V4q** raised concerns about the short (1-epoch) training horizon. This was not extended in the rebuttal.
* **Marginal Gains/Statistical Significance:** Reviewers **HhYG** and **Zk9q** remained concerned about the marginal nature of the improvements, which is exacerbated by the new baseline results.

**Reviewer Scores:**

* **Reviewer m6ay (6 -> 4):** While technical questions were answered, the two main weaknesses (limited evaluation setting/small models and incomplete baselines) were not effectively resolved. The new data showed the proposed method often performing worse than the requested baseline (DAPO).
* **Reviewer Zk9q (4 -> 4):** The reviewer explicitly questioned the "practical effectiveness". The rebuttal failed to provide results on 7B models as requested, and the comparison with DAPO showed inconsistent gains, failing to alleviate the reviewer's primary concerns.
* **Reviewer 5V4q (4 -> 4):** The reviewer's major concerns regarding experimental rigor (greedy decoding, 1-epoch training, lack of 7B models) were defended by the authors but not experimentally addressed. The fundamental concern about the evaluation protocol remains.
* **Reviewer HhYG (4 -> 4):** The reviewer was concerned about marginal improvements and lack of statistical significance. The rebuttal did not provide the requested significance analysis, and the new baseline results further suggest the gains may not be robust.

---

### Decision · Program_Chairs · 2026-01-26

Reject